# Development of a standard database of reference sites for validating global burned area products

Magí Franquesa[1], Melanie K. Vanderhoof[2], Dimitris Stavrakoudis[3], Ioannis Z. Gitas[3], Ekhi Roteta[4], Marc Padilla[5,6], Emilio Chuvieco[1]

[1]Environmental Remote Sensing Research Group, Department of Geology, Geography and the Environment, Universidad de Alcalá, Calle Colegios 2, Alcalá de Henares, 28801, Spain
[2]U.S. Geological Survey, Geosciences and Environmental Change Science Center, P.O. Box 25046, DFC, MS980, Denver, CO 80225, United States
[3]Laboratory of Forest Management and Remote Sensing, School of Forestry and Natural Environment, Aristotle University of
Thessaloniki, P.O. Box 248, GR-54124, Greece
[4]Department of Mining and Metallurgical Engineering and Materials Science, School of Engineering of Vitoria-Gasteiz, University of the Basque Country UPV/EHU, Nieves Cano 12, Vitoria-Gasteiz, 01006, Spain
[5]Centre for Landscape& Climate Research, Department of Geography, University of Leicester, Leicester LEI 17RH, United Kingdom
[6]Space Division, Starlab Barcelona, 47bis Tibidabo Avenue, 08035, Barcelona, Spain

Correspondence to: Magí Franquesa (magin.franquesa@uah.es)

**Abstract.** Over the past two decades, several global burned area products have been produced and released to the public. However, the accuracy assessment of such products largely depends on the availability of reliable reference data that currently
do not exist on a global scale or whose production require a high level of dedication of project resources. The important lack of reference data for the validation of burned area products is addressed in this paper. We provide the first publicly available Burned Area Reference Database (BARD) that was created by compiling existing reference BA datasets from different international projects. BARD contains a total of 2,661 reference files derived from Landsat and Sentinel-2 imagery. All those files have been checked for internal quality and are freely provided by the authors. To ensure database consistency, all files
were transformed to a common format and were properly documented by following metadata standards. The goal of generating this database was to facilitate BA algorithm developers and product testers reference information that would help to develop or validate new BA products. BARD is freely available at: https://doi.org/10.21950/BBQQU7 (Franquesa et al., 2020).

## 1 Introduction

Validation is defined by the Committee on Earth Observing Satellites Working Group on Calibration and Validation (CEOS-
WGCV) as "the process of assessing, by independent means, the quality of the data products derived from the system outputs" (CEOS-WGCV, 2012). Validation helps in evaluating the utility and limitations of using any remote sensing (RS) product, particularly on whether user accuracy requirements are met. For this reason, validation should be part of any RS project, even though it requires additional effort and cost that is not aimed at improving accuracy but rather to measure it. Validation implies

comparing our results to reference data, assumed to represent the actual conditions of the target variable at the satellite overpass time. In the case of global studies, it is very difficult to generate reference data for the wide variety of planetary conditions, thereby complicating validation. Some of the global variables (e.g. temperature and surface radiation) can be validated from ground sensor networks, such as weather stations, buoys or Aerosol Robotic NETwork (AERONET) sensors. Other variables are more difficult to validate, as they require generating global reference data that are based on higher-resolution sensors than those used to obtain the global product. This is the case of land cover or burned area products, which require first designing a sample strategy using statistically valid protocols and then extracting from the selected sites the reference polygons to be compared with the global datasets. Despite the time and effort required to derive reference datasets, accuracy assessment is a critical part of any global RS project and making these reference datasets publicly available will facilitate product comparison and lower the burden of validating future products.

Several global burned area (BA) products have been produced in the last two decades, providing an estimation of fire activity worldwide (Chuvieco et al., 2019). The first of these products was the Global Burned Area (GBA2000), based on daily VEGETATION (VGT, 1 km resolution) images acquired in the year 2000 and was generated by the Joint Research Centre of the European Union (Grégoire et al., 2003). The same year, the European Space Agency developed the GLOBSCAR BA product, also at 1 $km^2$, derived from daytime ERS-2 (European Remote Sensing Satellite) ATSR-2 (Along Track Scanning Radiometer) data (Simon et al., 2004). Other 1 km resolution global BA products released by European projects include the L3JRC (Tansey et al., 2008) covering the period from 2000 to 2007; GlobCarbon (Plummer et al., 2006), produced from 1998 to 2007; and the Copernicus GIO_GL1_BA products. These three products were derived from VGT images, although in the GlobCarbon project, ATSR images were used as well. More recently, the FireCCI (Climate Change Initiative) project (https://esa-fire-cci.org, last access: 25 March 2020), part of the European Space Agency (ESA) CCI programme, has generated three global BA products, based on Medium Resolution Imaging Spectrometer (MERIS) at 300m resolution (FireCCI41: Alonso-Canas and Chuvieco, 2015) and Moderate Resolution Imaging Spectroradiometer (MODIS) 250m data (FireCCI50: Chuvieco et al., 2018 and FireCCI51: Lizundia-Loiola et al., 2020). NASA (National Aeronautics Space Administration) released in mid-2008 the MCD45A1 product derived from 500 m MODIS imagery (Roy et al., 2008), which has now been superseded by MCD64A1 at the same resolution but with a different BA algorithm approach (Giglio et al., 2009; 2018).

These global BA products have been validated by comparing them with reference data generated from medium resolution sensors (such as those on board the Landsat, SPOT (Satellite Pour l'Observation de la Terre), or Sentinel-2 missions). These reference data were typically derived from multitemporal pairs of images to properly date the validation period.

According to the representativeness of samples used to perform product validation, the CEOS-WGCV Land Product Validation (LPV) subgroup defined four validation stages with the level of sampling effort and statistical rigor increasing at each stage (https://lpvs.gsfc.nasa.gov/, last access: 25 March 2020). Early validation exercises were subjected to a first stage validation, usually based on small samples of reference sites that were not selected using a probability sampling design, but rather by a purposeful or convenience selection based on data availability or expert knowledge to ensure diverse wildfire conditions were included in the sample (Tansey et al., 2004; Roy et al., 2005). Roy and Boschetti (2009), for instance, reported validation

results for the MCD45A1 product using a set of 11 Landsat scenes distributed across southern Africa. Chuvieco et al. (2008) validated a regional product for Latin America using 19 Landsat scenes and 9 China–Brazil Earth Resources Satellite (CBERS) scenes that were donated by regional space agencies when access to the Landsat archive was not yet free and open to the public, thereby limiting the number of selected validation sites. The MCD64A1 Collection 5 was not formally validated, and the most recent MCD64A1 Collection 6 products were first validated using a set of 108 Landsat scenes distributed across a wide range of fire-affected ecosystems but not selected via probability sampling (Giglio et al., 2018). A recent study has provided a validation of the MCD64A1 product implementing a probability sampling design and using Landsat-8 Operational Land Imager (OLI) images, but only for a single year (Boschetti et al., 2019). Previous statistical validation of NASA and FireCCI BA products were conducted by Padilla et al. (2014; 2015) using a set of 105 randomly selected Landsat scenes for a single year (2008) and by Chuvieco et al. (2018) using a multitemporal reference dataset of 12 years. Other projects covering large areas have been developed in the USA using Landsat data across six years (Vanderhoof et al., 2017) and Africa using Sentinel-2 Multispectral Instrument (MSI) images (Roteta et al., 2019) where validation sites were selected through probability sampling. In all cases, reference datasets were created based on independent interpretation of BA, controlled by visual inspection. The importance of applying probability sampling to collect reference data has been highlighted by different authors as a critical feature of the sampling design protocol to achieve statistically rigorous assessment (Stehman, 2001; 2009; Olofsson et al., 2014; Stehman and Foody, 2019). Thus, in contrast to such reference data collected by convenience, ease of access, or other methods that lack randomization, data collected through probability sampling makes it possible to obtain rigorous estimates of accuracy.

The main bottleneck for validating global BA products or global BA algorithms is the generation of reference BA datasets. To facilitate the activity of BA algorithm developers, this paper aims to present and deliver to the scientific community the Burned Area Reference Database (BARD), a set of reference BA perimeters that can be used as reference data for validation of BA products or to help the development of BA algorithms (obviously, the same files cannot be used for both training and validating an algorithm). These validation files were compiled from different international projects and years, therefore the resulting database will facilitate the assessment of BA algorithms in a wide range of ground conditions.

The BARD includes the following datasets of reference data: FireCCI global (2008), FireCCI global (2003-2014), FireCCI Africa (2016), FireCCI Africa S2 (2016) that were produced within the framework of the FireCCI project; the CONUS (contiguous United States) Landsat Burned Area (1988-2013), developed within the Landsat Level-3 Science Products project, and NOFFi Greece (National Observatory of Forest Fires, 2016-2018) that was produced within the NOFFi project.

The paper presents the methods that were used to generate the BA reference data paying particular attention to the sampling design and reference data retrieval methods applied to the different BARD datasets. The data specifications to transform all the files to a common standard format and file structure are then presented. Finally, a detailed description of each dataset included in BARD is provided and the main dataset features are then summarized to facilitate a general overview.

## 2 Methods

### 2.1 Selection of validation sites: sampling design

High-quality reference data generation is an expensive and time-consuming task, which constrains the total number of validation sites that can be established in any validation exercise. For this reason, sampling design is critical to make the most of the resources available and ensure the highest precision of accuracy estimates given the available resources to generate reference data. Padilla et al. (2014; 2015) implemented a stratified random sampling design that allowed for global BA accuracy inferences for the first time. Boschetti et al. (2016) extended the sampling design to include the temporal dimension of the sampling units. More recently, Padilla et al. (2017) presented a first approach to efficiently stratify the population and allocate the samples across strata. Chuvieco et al. (2018) conducted a multi-annual accuracy assessment across 12 calendar years (2003-2014), reporting for the first time the temporal accuracy variation of global BA products. Meanwhile, Boschetti et al. (2019) validated the MCD64 c6 BA product, but instead of using the calendar year, the authors used a fire year (from March 1$^{st}$ 2014 to March 19$^{th}$ 2015) as defined in Boschetti and Roy (2008).

The sampling design protocols to validate BA products were therefore developed considering the rarity and ephemeral nature of the BA, which is indeed a special case of land-cover change (Stehman and Foody, 2019). When selecting samples for obtaining probability inferences, the allocation of samples should follow a probability sampling design, to compute unbiased population estimates. For BA product validation, this implies selecting samples considering the spatial and temporal dimension. The spatial dimension of sampling units is usually defined by the Thiessen scene areas (TSAs) constructed by Cohen et al. (2010) and Kennedy et al. (2010) specifically for use with Landsat WRS-2 frames (Worldwide Reference System, Fig. 1a). The key advantage of TSAs is that they provide non-overlapping Landsat-like frames, which allow for a convenient computation of unbiased estimators (Gallego, 2005). The temporal dimension of sample units is defined by the acquisition dates of the pre- and post-fire images. For example, in Boschetti et al. (2019), the validation period (1 year) was divided into equal temporal size sampling units using the 16-day Landsat 8 acquisition interval, thus allowing for the temporal random selection of the reference images. This temporal partitioning, also makes it possible to intensify the sample in strata that comprise the fire season and where burning is more likely to occur (Stehman and Foody, 2019). However, longer period intervals (>100 days) are used to define sampling units to allow a long temporal overlap of reference data with the BA product, which helps to disentangle the spatial errors from the temporal errors of the BA product (Roteta et al., 2019; Lizundia-Loiola et al., 2020).

In any case, sample units are then stratified to properly represent the variety of conditions that affect the accuracy of BA products. This stratification is usually based on (a) major Olson biomes (Olson et al., 2001) (Fig. 1b) and (b) the BA extent provided by a global BA product considered to be reliable or active fire detections, assigning each sample unit to high or low BA strata based on a threshold that can be specifically adapted to each biome stratum as in Padilla et al. (2017) or simply set as the 20$^{th}$ quantile of the cumulative distribution of active fire counts as in Boschetti et al. (2016; 2019).

One of the advantages of the stratified sampling design adopted for BA maps validation previously mentioned was that it allows for rigorous estimates of global BA accuracy. However, another key advantage of stratified random sampling design that should be strongly emphasized is that it makes it possible to increase the sample size of an initial global sample for specific regions or rare land-cover classes (Stehman et al., 2012). This is the case of the CONUS Landsat Burned Area (1988-2013) dataset where reference sites for the CONUS extent were augmented based on the initial sample of the FireCCI global (2008) dataset.

Stratified random sampling design was applied to several datasets included in BARD: FireCCI global (2008), FireCCI global (2003-2014), FireCCI Africa (2016) and the CONUS Landsat Burned Area (1988-2013). FireCCI Africa S2 (2016) was obtained also by probability sampling but, in this case, applying a systematic sampling design. NOFFi Greece (2016-2018) is the only dataset of BARD that was obtained through convenience sampling rather than probability sampling.

To report BA accuracy from these stratified sample datasets, users should apply the proper estimation formulas detailed in the associated articles (see Table 2) and use the additional information as the stratum of each sampled unit and the stratum sizes of the stratified sampling, provided in the metadata files and tables of appendix A, respectively.

## 2.2 Reference data generation methods

Following the recommendations of the CEOS Calibration/Validation group, all the burn perimeters of BARD were derived from multitemporal comparison of medium resolution satellite imagery (Landsat TM (Thematic Mapper)/ETM+ (Enhanced Thematic Mapper plus)/OLI or Sentinel-2 MSI). Burned patches included in the files are only those that occurred in between the two satellite images used to generate the reference data (Fig. 2). The procedures implemented to obtain those burned patches are diverse, depending on the dataset, but all include a semi-automatic procedure (e.g. Bastarrika et al., 2011) and then a visual inspection to confirm that the detected perimeters were actually burned areas. In some cases, the semiautomatic classification was enhanced with polygons manually digitized. In several cases, this visual inspection was confirmed by another interpreter to double check the quality. When parts of the scene could not be observed or interpreted because of clouds or sensor problems (i.e. Scan Line Corrector (SLC)-off problems of ETM+), either in the pre- or post-fire images, they were classified as no-data. This was done to make sure that only areas with reliable data were included in the reference files. Regarding 'unburned' category of reference data, different criteria were applied to label seas and inland water bodies in the different datasets. Thus, for FireCCI global (2008), FireCCI global (2003-2014), FireCCI Africa (2016) and CONUS Landsat Burned Area (1988-2013) datasets, surface waters were classified as 'unburned' while in FireCCI Africa S2 (2016) and NOFFi Greece (2016-2018), the 'no-data' category was applied to label them.

It should be noted that reference data are not just high accuracy BA products generated by well-designed algorithms using medium- or high-resolution imagery. Rather, reference data following international standards should provide reliable burned area but also the unburned surface of the interpreted geographic region and the unobserved/unmapped areas within the region, as shown in Fig. 2c.

Like the sampling units from which reference data are derived, reference data can be defined by its spatial and temporal dimension. The spatial dimension is a function of the geographic extent interpreted to obtain the reference data, where the size varies depending on the criteria adopted in each project. For example, reference data from the FireCCI global (2003-2014) dataset were spatially defined by a frame of 30 x 20 km located at the centre of the Landsat images, whereas the entire Landsat scenes were used in the case of the CONUS Landsat Burned Area (1988-2013) dataset. The spatial extent used in the datasets included in BARD will be specified in section 2.4 where a detailed description of each dataset is provided.

The temporal dimension of the reference data represents the period defined by the acquisition date of the pre- and post-fire images used to generate them. Regarding the temporal length of the reference data, the FireCCI project adopted the terms 'short unit' (SU) and 'long unit' (LU). The former refers to those reference data derived from a pair of consecutive images separated by 16 days or less (the temporal span between two Landsat acquisitions). The latter is defined by a series of consecutive SUs covering at least 100 days. LUs allow for long temporal overlaps between validation and product data, reducing or minimizing the impact of the product's temporal reporting accuracy in the accuracy estimates (Padilla et al., 2018). The combined use of SUs and LUs is useful to assess such and contextualize impact (Lizundia-Loiola et al., 2020). A LU BA map consists in the combination of consecutive SU maps (Fig. 3). A pixel classified as no-data in any of the SU maps is kept as such in the LU BA map. This is to ensure that any pixel available data is observed frequently (every 16 days or less) and an eventual burn is not missed due to simply a fast recovery of the vegetation. The permanently observed pixels, were classified as burned in the LU if they were detected as burned in any SU of the time series covered by the LU. The presence of no-data (e.g. due to clouds) in a single image may reduce drastically the spatial cover of available data in the resulting LU. Therefore, BA maps are generated for every single SU, but the BA map for a LU is generated by accumulating the consecutive SUs of the same TSA. The length of the LU would depend on the existing cloud-free consecutive SUs. For example, if 8 consecutive SUs, all covering the same temporal length (e.g. 16 days) are cloud free and the 9th image has 90% of the area cloud covered, the LU would include only the first 8 SU maps, even if SU were generated for the 9th and 10th consecutive images.

As burning is detected on any given single image in between the period covered by two satellite acquisitions, all burned patches are dated based on the second reference image of a multitemporal pair. Therefore, SUs will have the same date for all the burned patches, while LU reference data will have burned patches from different dates as multiple pairs of images are used to build the LU (Fig. 3).

Among the datasets included in BARD, SUs were used in the FireCCI global (2003-2014) dataset as part of the sampling design, and LUs were used for the FireCCI Africa (2016) dataset. Reference data from the rest of the FireCCI project datasets (FireCCI global (2008) and FireCCI Africa S2 (2016)) and CONUS Landsat Burned Area (1988-2013) dataset, were retrieved from a single pair of images with a variable time lapse between pre- and post-fire images. Thus, the temporal length of those reference data was determined by the availability of suitable images and the duration of the burned signal. The NOFFi Greece (2016-2018) reference data were obtained considering a time-series of Sentinel-2 images, but with variable length and non-consecutive time-series step.

## 2.3 Data specifications

Each dataset of BARD is organised in three folders with associated files including: (a) 'metadata', which contains a .csv file containing the file name of all the reference files included in the dataset, along with additional information such as the temporal length (days), the total number of images interpreted (n_images), the area (m$^2$) of each mapped category ('burned', 'unburned' and 'unobserved'), the land surface and total area of each reference data file. For those datasets where a stratified random sampling design was used, the .csv file also specifies the stratum of each sampled unit and the size (tsa_area) of the corresponding TSA; (b) 'regions', which contains an ESRI shapefile (*.shp) containing all the sample sites (TSAs or Sentinel-2 tiles) covered by the dataset; and (c) 'shapefiles', containing the validation reference shapefiles ordered by year. They are also released in shape (.shp) format.

All datasets are in UTM/WGS84 projection. The name of the files is defined as follows: 'Project_RD_ppprrr_yyyymmdd_yyyymmdd' (e.g. FireCCI_RD_164069_20160514_20160709'), where:

Project = Project in which the reference data were generated.

RD = stands for Reference Data.

ppprrr = refers to the Landsat Worldwide Reference System (WRS) path (ppp) and row (rrr) of the scene. For collections where Sentinel-2 was used instead of Landsat images, ppprrr refers to the Sentinel-2 tile (e.g. FireCCI_RD_T28PET_ 20160111_20160311').

yyyymmdd (year, month, day). The first date corresponds to the pre-fire date, which is the date of the first image used for BA detection; the second one refers to the post-fire date, which is the date of the last image used for generating the reference fire perimeters.

The following attribute fields are included in the shapefiles (Table 1):

- category:
  - 1: Burned area. This category includes all polygons detected as burned
  - 2: No-Data. This category includes all polygons that could not be interpreted or were not observed by the sensor, either by clouds and/or cloud shadows, topographic shadows, smoke, or sensor errors (for instance, those caused by SLC-off problems of ETM+ after May 31, 2003).
  - 3: Unburned. This category includes all polygons observed as not burned within the limits of the area covered by the image.
- preDate: Acquisition date of the image taken before the occurrence of the fire: yyyy-mm-dd (year, month, day).
- postDate: Acquisition date of the image taken after the fire: yyyy-mm-dd (year, month, day).
- preImg and postImg: The pre- and post-fire Landsat scene identifier (e.g. 'LC80260422013124LGN01'). For reference files based on S2 images, the datastrip ID is used instead.
(e.g. 'S2A_OPER_MSI_L1C_TL_SGS__20160420T171415_A004324_T28PEB_N02.01').

- path: The Worldwide Reference System-2 (WRS-2) path of the Landsat scene. For reference files based on S2, the tile number was used.
- row: The row of the Landsat scene. For reference files based on S2, the tile number was used.
- year: The year of the validation dataset.
- area: Area in square meters (m$^2$) calculated on the WGS84/UTM Cartesian plane.

## 2.4 Reference datasets

### 2.4.1 FireCCI global (2008)

The FireCCI global 2008 reference dataset was created using a stratified random sampling design ((Padilla et al., 2014; 2015), Table A1). Two levels of spatial stratification were used to select the spatial units based on TSAs derived from the Landsat World Reference System 2 (WRS-2). Spatial units were first stratified across seven aggregated Olson biomes (Olson et al., 2001). Each biome was stratified into high and low BA extent based on the Global Fire Emissions Database (GFED) Version 3 (Giglio et al., 2009; 2010). A total of 101 images from Landsat-5 TM and 109 for Landsat-7 ETM+ satellite sensors were used to retrieve BA perimeters. The complete scene was used for Landsat-5 TM images, whereas only the centre of Landsat-7 ETM+ scenes were interpreted in order to avoid data SLC gaps. BA perimeters were derived using a semi-automatic algorithm developed by Bastarrika et al. (2011), where high burn severity pixels were selected to train core burned area, and adjacent lower burn severity pixels were added to the core detected patches using a region-growing algorithm. The FireCCI global 2008 dataset includes 105 reference data files, derived from single pair of images, for the year 2008. The temporal length of reference data varies between 8 and 144 days: 79% of image pairs were separated by 32 days or less, 16% between 32 and 100 days, and 5% by more than 100 days with a maximum time gap between the pre- and post-fire image of 144 days. The total area of reference data is 1.76·10$^6$ km$^2$, of which 1.35% corresponds to burned category, 88.35% to unburned and 10.30% to unobserved category. The location and temporal length of the reference data is shown in Fig. 4. This reference dataset is compliant with CEOS-LPVS Stage 3.

### 2.4.2 FireCCI global (2003-2014)

The FireCCI global (2003-2014) dataset covers a period of 12 years, from 2003 to 2014 (Padilla et al., 2018), and was generated in the framework of the FireCCI project with the collaboration of the Copernicus Global Land Service (CGLS). The reference data were derived from consecutive Landsat images separated by 8-16 days for each selected TSA and year. A total of 585 images from Landsat-5 TM, 1564 from Landsat-7 ETM+ and 209 from Landsat-8 OLI satellite sensors, were used to retrieve BA perimeters. The sampling units were selected following a stratified random sampling design (Table A2). The total population of sample units were defined spatially by TSAs and temporally by the dates of Landsat images available, filtering out those with a cloud cover greater than 30%. For each calendar year, the sample units were stratified by Olson biomes (Olson et al., 2001) and BA based on MCD64A1 (Giglio et al., 2009). The threshold used to assign the high/low BA strata was defined

separately for each year and biome. Once the strata were defined by year-biome-BA, a set of 100 sampling units were selected for each calendar year applying a sample allocation according to Eq. (1):

$$n_h \propto N_h \overline{BA}_h \qquad\qquad (1)$$

where $n_h$ is the sample size to be selected in stratum $h$, $N_h$ is the stratum size and $\overline{BA}_h$ the BA mean in stratum $h$.

Finally, a spatial subset window of 30 x 20 km located at the centre of the images was applied for interpretation and BA

reference data retrieval. The reference perimeters were extracted from a dedicated Random Forest algorithm, trained for each sampling site, and output maps were visually inspected by two interpreters (Padilla et al., 2018).

The FireCCI global (2003-2014) dataset includes 1200 reference data files from 722 different TSAs and 12 years, from 2003 to 2014. The temporal length of reference data varies between 8 and 16 days. The total area of reference data is $0.72 \cdot 10^6$ km$^2$, of which 3.85% corresponds to burned category, 71.85% to unburned, and 24.29% to unobserved category. The location and

total number of reference data in each TSA are shown in Fig. 5. This reference dataset is compliant with CEOS-LPVS Stage 3.

### 2.4.3 FireCCI Africa (2016)

The FireCCI Africa reference dataset consists of LU BA maps and was generated for the year 2016 from Landsat imagery (Padilla et al., 2018). It was also generated in the framework of the FireCCI project with the collaboration of the CGLS. The

sampling was designed with long units and it was similar to that for the FireCCI global (2003-2014) dataset, as mentioned in the previous section (Table A3). The only difference was the sample size, 50 units instead of 100 units per year. Note that each unit here is much larger, as it consists of multiple image pairs. Two reference perimeter datasets are released: (a) Reference data at SU level, 1052 files with 8-16 day BA maps; and (b) Reference data at LU level, 50 files. The temporal length covered at each LU varies from 24 to 256 days (Fig. 6b): 18% of the LUs cover a temporal length below 50 days, 34% between 50 and

100 days, and 48% are above 100 days. As mentioned in Section 2.2., LUs were defined to be at least 100 days long, although the presence of clouds reduced the actual temporal periods with available data. The total area of LU reference data is $0.023 \cdot 10^6$ km$^2$, of which 15.72% corresponds to burned category, 49.61% to unburned, and 34.67% to unobserved category. The location, number of image pairs, and temporal length of the LUs reference data are shown in Fig. 6. This reference dataset is compliant with CEOS-LPVS Stage 3.

### 2.4.4 FireCCI Africa S2 (2016)

The FireCCI Africa S2 BA reference dataset was created to perform an initial validation assessment of the Small Fire Database Fire_cci v1.1 product (FireCCISFD11) produced for the year 2016 for the whole Sub-Saharan Africa (Roteta et al., 2019). Reference data were generated from the comparison of two Sentinel-2 MSI images at 20 m resolution per reference site. Systematic sampling was used to select 52 validation sites based on Sentinel-2 tiles (110 x 110 km) over Sub-Saharan Africa.

BA was mapped with the BAMS methodology, which is a semi-automated algorithm (Bastarrika et al., 2014). In short, training polygons for the burned category were defined in each tile, and burned seeds were detected. Then, burned pixels were grown

out from these seeds until all pixels for each burned patches were detected. The results were visually analysed to determine the accuracy of the classification and new training polygons were defined if needed. This was done sequentially until all burned areas were mapped and no commission or omission errors were visually detected. Finally, if there was noise created by unmasked clouds and cloud shadows, it was edited and removed manually.

The temporal length of the reference data varies between 10 and 120 days: 86% of the pairs of images were separated by less than 50 days and 14% by more than 50 days with a maximum time lapse of 120 days. The total area of reference data is $0.63 \cdot 10^6$ km$^2$, of which 8.87% corresponds to burned category, 72.42% to unburned, and 18.71% to unobserved category. The location and temporal length of the reference data are shown in Fig. 7. This reference dataset is compliant with CEOS-LPVS Stage 1.

### 2.4.5 CONUS Landsat Burned Area (1988-2013)

CONUS Landsat Burned Area (1988-2013) reference dataset (Vanderhoof et al., 2017; 2020) extends across the contiguous United States (CONUS) and was generated to validate the Landsat Burned Area product (Hawbaker et al., 2017; 2020). The sampling design was adapted from the methods used by the ESACCI FireCCI project. Existing FireCCI validation TSAs (n=9) within CONUS were augmented with an additional 19 TSAs for a total of 28 TSAs. The TSAs were stratified across the major Olson biomes (Olson et al., 2001) including (1) temperate forest, (2) Mediterranean forest, (3) temperate grassland and savannah, (4) tropical and subtropical grasslands and savannah, and (5) xeric/desert shrub. TSAs selected within each biome were meant to represent high and low burned areas as specified by the Global Fire Emissions Database (GFED) version 3 (Table A4). Systematic sampling was applied to select 6 validation years spaced out in 5-year increments (2013, 2008, 2003, 1998, 1993 and 1988).

A total of 269 images from Landsat-5 TM, 10 from Landsat-7 ETM+, and 56 from Landsat-8 OLI were used to derive the BA extent. Landsat reference images were limited to those with a geometric Root Mean Square Error (RMSE) < 10 m, <20% cloud cover, and available as a L1T Surface Reflectance product. Time lapse between images was not limited to 16 days and only two images (pre- and post-fire) were used to retrieve BA reference data for each validation site and year. The pre- and post-fire image pairs did not specifically represent a probability sample within a year but were designed to target changes incurred over the peak fire season. Peak fire season was determined using the distribution of total burned area by month as derived from the MCD45 burned area product (2001-2015). The FMask from the Landsat surface reflectance product was applied to mask clouds, cloud shadows, snow and open water from each image used (Zhu and Woodcock, 2014). For Landsat-7 ETM+ images, SLC off pixels were masked. The low-, medium- and high-intensity development classes (i.e. urban areas) were masked using the National Land Cover Database (NLCD, https://www.mrlc.gov/national-land-cover-database-nlcd-2016) (Homer et al., 2015) to reduce spectral confusion between burned areas and impervious surfaces. Similarly, agricultural burns were not used to train the reference data burn classification, therefore the accuracy of the reference dataset in agricultural areas is unknown. If this is of concern to users, then users can mask the 'cultivated crops' land cover type from the reference data using the NLCD.

Burned area maps were generated using BAMS (Bastarrika et al., 2014). The Normalized Burn Ratio (NBR), Mid-infrared Burned Index (MIRBI), Global Environmental Monitoring Index (GEMI) and Normalized Difference Vegetation Index (NDVI) were calculated for the pre- and post-fire images and utilized in a supervised classification. The algorithm was trained on manually selected polygons containing (1) clearly burned pixels and (2) spectrally similar but less distinct burned pixels. The algorithm applied a region-growing function between the two types of training polygons, while cut-off values for each

variable were extracted from the training polygons. Each classified burned area was then manually edited. When available, the analysts utilized ancillary datasets (e.g. Monitoring Trends in Burn Severity (MTBS, Eidenshink et al., 2007), MODIS active fire points (MOD14 collection 5, Giglio et al., 2009), MODIS burned area (MCD45A1 collection 5, Roy et al., 2008), and aerial imagery) to improve the confidence in their selection of training pixels and manual edits. To maximize the accuracy of the reference dataset, each image pair was classified into burned area extent and visually evaluated and edited independently

by three different analysts. A pixel was then classified as burned if it was identified as burned by two of the three analysts. Additional processing details can be found in Vanderhoof et al. (2017).

The CONUS Landsat Burned Area (1988-2013) dataset includes 168 reference data files from 28 Landsat path/rows and six years (1988, 1993, 1998, 2003, 2008, 2013). The temporal length of reference data varies between 16 and 288 days: 37% of pairs of images were separated by less than 50 days, 35% between 50 and 100 days, and 28% by more than 100 days with a

340 maximum time lapse between the pre- and post-fire image of 288 days. The total area of reference data is $5.23 \cdot 10^6$ km$^2$, of which 0.12% corresponds to burned category, 82.33% to unburned, and 17.55% to unobserved category. Location of reference sites based on TSAs is shown in Fig. 8. With the publication of Hawbaker et al. (2020), this reference dataset is compliant with CEOS-LPVS Stage 4.

### 2.4.6 NOFFi Greece (2016-2018)

The reference data were obtained using the perimeters produced by the National Observatory of Forest Fires (NOFFi) (http://epadap.web.auth.gr, last access: 25 March 2020) and, specifically, its Object-based Burned Area Mapping (OBAM) service, implemented by the Laboratory of Forest Management and Remote Sensing (FMRS) of the Aristotle University of Thessaloniki. NOFFi-OBAM is an on-demand service, meaning that it is activated after large wildfire events and under explicit requests by the local forest offices. It relies solely on Sentinel-2 imagery and is employed only for fires within Greece. The

NOFFi-OBAM algorithm is designed to map fire perimeters and follows a supervised learning approach using a post-fire Sentinel-2 (Level-1C) image, although a pre-fire image is also used for photo-interpretation purposes. The methodology applied to retrieve the fire perimeters is fully described in Tompoulidou et al. (2016). Non-probability sampling design was applied for this dataset; reference sites were selected by convenience based on images previously processed in the NOFFi-OBAM service.

The NOFFi-OBAM fire perimeters were used as the basis for creating the reference data for the NOFFi Greece reference dataset considering the burned area mapping years 2016, 2017 and 2018. For each Sentinel-2 tile ID (e.g. T34SDH) in which fire perimeters were available, the whole time-series of images were visually checked and the date range for the reference file

creation was defined from the first pre-fire image to the last post-fire image. Small fires within the specific time series that were not mapped from the NOFFi-OBAM service were explicitly digitized. Since NOFFi-OBAM only serves Greece, areas outside Greece's official land boundaries (e.g. seas and land areas of neighboring countries) were masked and classified as unobserved surfaces (category = 2). Some burned scars in overlapping border tiles were mapped by using images from those neighboring tiles only if the post-fire image used for the mapping was inside the time span of the former tile ID. For example, the file 'NOFFi_RD_T34SGH_20160710_20160730.shp', includes polygons with preImg/postImg from T35SCK. This can be identified from the preImg, postImg, and tile columns of the file. Clouds and cloud shadows were manually digitized and masked (category = 2), considering the last postImg. Although a non-probability sampling design was applied for this dataset, the NOFFi-OBAM service has been activated for all wildfires greater than 100 ha during the period 2016–2018 and, in many cases, for smaller (or even much smaller) wildfires. Therefore, the dataset contains a representative set of Sentinel-2 tiles that are frequently affected by wildfires in Greece, at least for the given time-period.

The NOFFi Greece dataset includes 34 reference data files from 25 different Sentinel-2 tiles. The temporal length of reference data varies between 5 and 132 days. The total area of reference data is $0.41 \cdot 10^6$ km$^2$, of which 0.10% corresponds to burned category, 25.83% to unburned, and 74.08% to unobserved category. As shown in Fig. 9, most of the surface of the tiles from this dataset corresponds to sea surface that was labelled as 'no-data' (section 2.2.), this is the reason the unobserved category is so high compared to the rest of the datasets. The location and temporal length of the reference data as well as the number of images used in each reference site are shown Fig. 9. This reference dataset is compliant with CEOS-LPVS Stage 1.

## 3 Data availability

The BARD compiled in this effort is freely available on the e-cienciaDatos repository (https://doi.org/10.21950/BBQQU7 (Franquesa et al., 2020)). All burned area reference data files have been visually checked, reprojected and reformatted to provide a uniform set of attributes and metadata descriptions to maximize the ease with which these reference files can be used to evaluate global burned area products. A summary of the data included in each dataset is described in Table 2 and 3. Reference shapefiles and metadata files can be downloaded grouped by the datasets described in this publication: FireCCI global (2008), FireCCI global (2003-2014), FireCCI Africa (2016), FireCCI Africa S2 (2016), CONUS Landsat Burned Area (1988-2013), and NOFFi Greece (2016-2018). Plans are underway to expand the Burned Area Reference Database with new reference files that the FireCCI project produces, and we encourage future contributions from the scientific community.

## 4 Conclusions

BARD is the first publicly available database that compiles and standardizes previously generated validation reference data. Reference datasets included in this database were produced throughout the life of the FireCCI project since 2010, and other initiatives as Landsat Level-3 Science Products and NOFFi projects have joined and contributed to this effort. BARD gathers

and compiles a total of 2661 standardized shapefiles representing reference burned area data generated from approximately 4500 Landsat and Sentinel-2 images and 8 million square kilometres of interpreted land surface. Reference data were produced following the recommendations of the CEOS Calibration/Validation group and visually inspected by two or more experienced interpreters to ensure the accuracy of the data. As BARD is a compilation of datasets that were produced in different projects and years in which different methods were applied (e.g. different sampling methods, sensors, years or region extent), it is highly recommended that the user clearly understands the characteristics of the dataset or datasets that best suits their needs. BA reference database and future updates remedy the lack of an extensive global and regional, multitemporal validation dataset (Humber et al., 2019) and, certainly, can serve as a valuable source for validation of existing products and developing new BA algorithms, particularly those requiring large amounts of training data.

## 5 Appendix A: Supplementary tables

## 6 Author contributions

MF and EC wrote the first draft of the manuscript. MF has coordinated the manuscript production and prepared the figures, standardized the reference files and organized the BARD database, and managed its publication on the e-cienciaDatos repository. MV provided the CONUS Landsat Burned Area (1988-2013) dataset. DS and IZG provided the NOFFi Greece (2016-2018) dataset. ER provided the FireCCI Africa S2 (2016) dataset, and MP provided the rest of the FireCCI datasets. EC, as the Science Leader of the FireCCI project, managed the overall execution of the project and suggested the preparation of the present article. All the authors have contributed to the writing and reviewing of the manuscript and agreed on the final version.

## 7 Competing interests

The authors declare that they have no conflict of interest.

## 8 Acknowledgements.

This research has been funded by the FireCCI project (contract no 4000126706/19/I-NB) which is part of the ESA Climate Change Initiative. We thank Joshua J. Picotte (U.S. Geological Survey Earth Resources Observation and Science (EROS) Center, USA), M. Lucrecia Pettinari (University of Alcalá, Spain), Renata Libonati, Julia A. Rodrigues (Universidade Federal do Rio de Janeiro, Rio de Janeiro, Brazil), and Alberto W. Setzer (National Institute for Space Research (INPE), Brazil) for their valuable suggestions in the first version of the manuscript. M. Vanderhoof's time was supported by the U.S. Geological

Survey, Land Resources Mission Area, Land Change Science Program. Any use of trade, firm, or product names is for descriptive purposes only and does not imply endorsement by the U.S. Government.

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

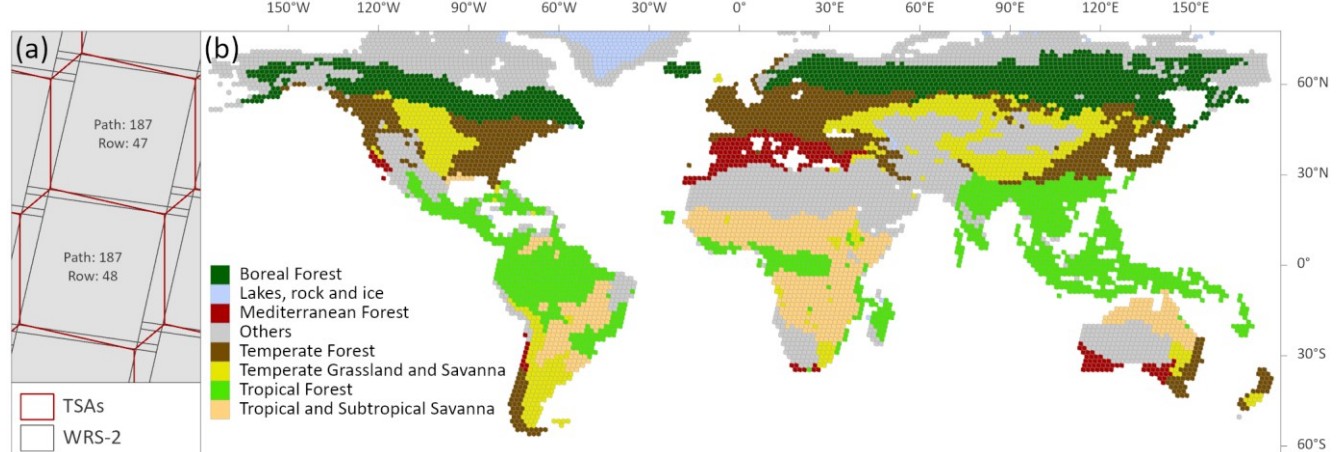

**Figure 1: (a)Thiessen scene areas (TSAs) based on Landsat Worldwide Reference System-2 (WRS-2) frames. TSAs are used as non-overlapping spatial units in the sampling design. (b) Distribution of major Olson biomes reclassified as in Padilla et al. (2014).**


**Figure 2: Example of Landsat-7 pre-fire (a) RGB (7,4,3) image and Landsat-8 post-fire (b) RGB (7,5,4) image. Both (a, b), were used to derive the 'FireCCI_RD_169065_20140712_20140720' BA reference file (c) at WRS-2 Landsat 169-065 path-row (East Africa).**

Time period between both images is 8 days: from 12 June to 20 June 2014. Only the land surface that burns between the two dates is classified as burned, while burned scars in the pre-fire image are assigned to the unburned category. Unobserved pixels on either 550 pre- or post-fire image due to the presence of clouds, cloud-shadows, SLC-gaps or smoke plumes are classified as no-data.

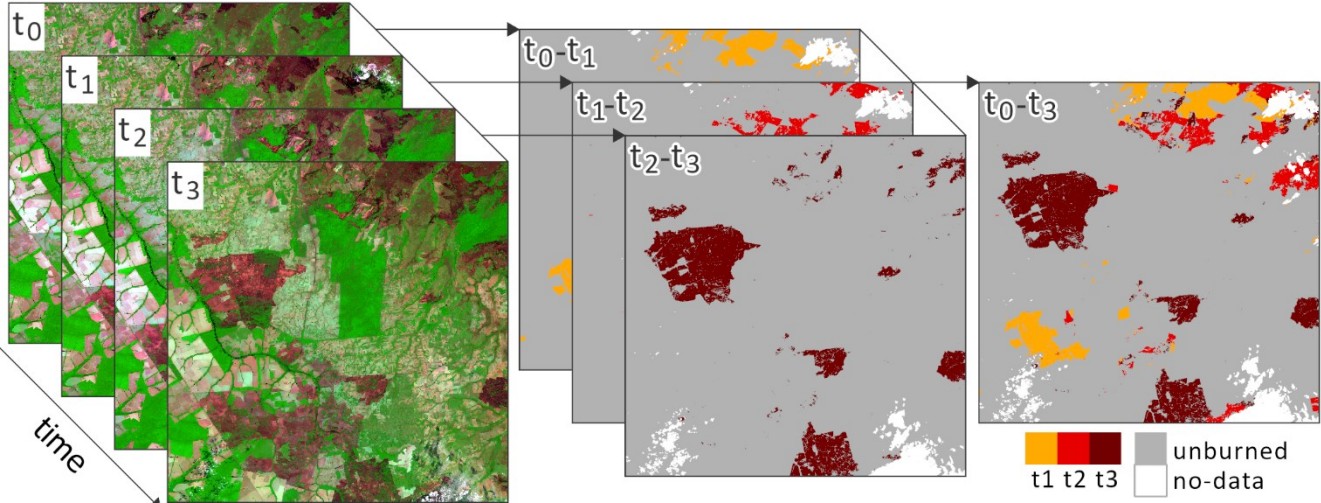

**Figure 3: Schematic process of long unit reference data generation. Consecutive image pairs are selected from the multitemporal image series at same location (left: Landsat-8 RGB (7,5,4) images time series) to derive the correspondent short unit reference data files (e.g. Image $t_0$ and $t_1$ to obtain the reference data $t_0$-$t_1$). From the union of the different short units we generate the long unit**
**reference data (right). The long unit $t_0$-$t_3$ includes all the burned scars that occurred between the first image ($t_0$) and the last image interpreted ($t_3$), burned scars from the first image ($t_0$) are not included or mapped. Unobserved areas in any of the images are labeled as no-data in the final long unit reference data. Colours (orange-$t_1$, red-$t_2$, brown-$t_3$) represent the dates in which the burned area patches were observed.**

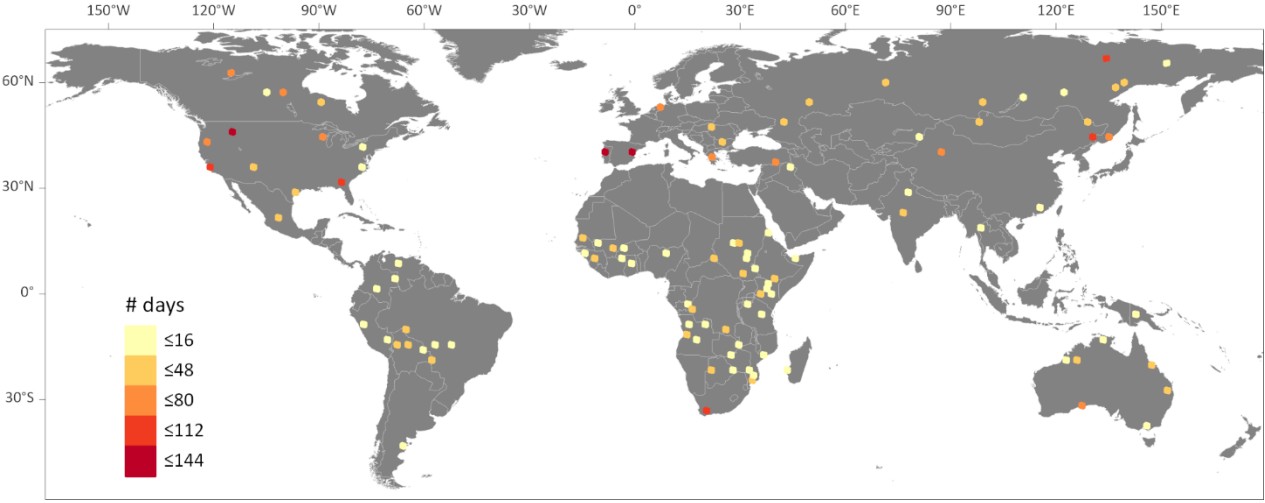


**Figure 4: Spatial distribution of the reference sites for FireCCI global (2008) dataset. The legend shows the temporal distance (days) between the pre- and post-fire images used in each validation site for the year 2008.**

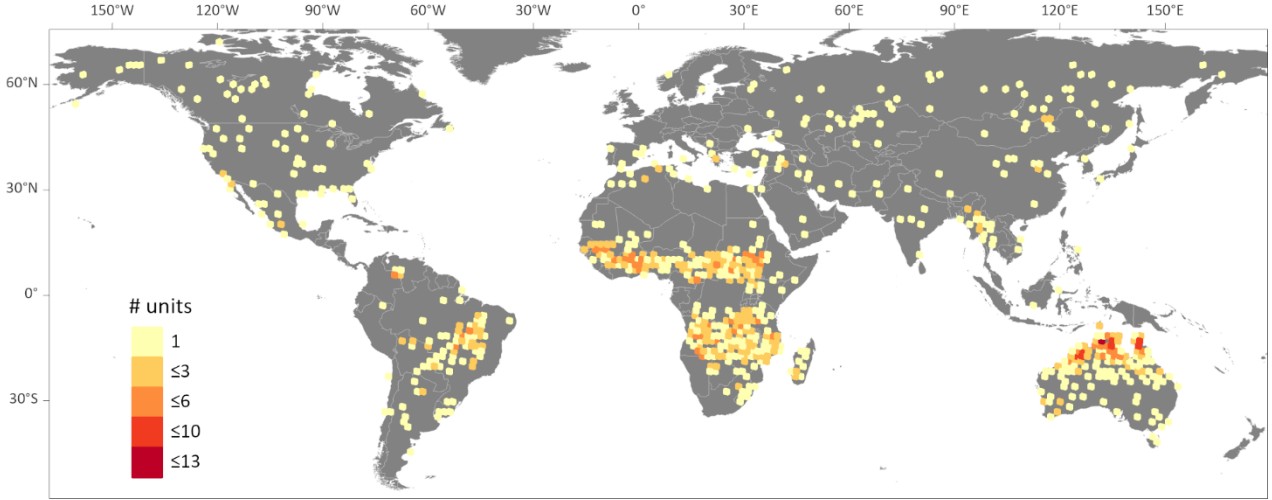

**Figure 5: Spatial distribution of the validation Thiessen scene areas (TSAs) for FireCCI global (2003-2014) dataset. The legend shows the total number of reference data files generated for each TSA between the period 2003-2014.**

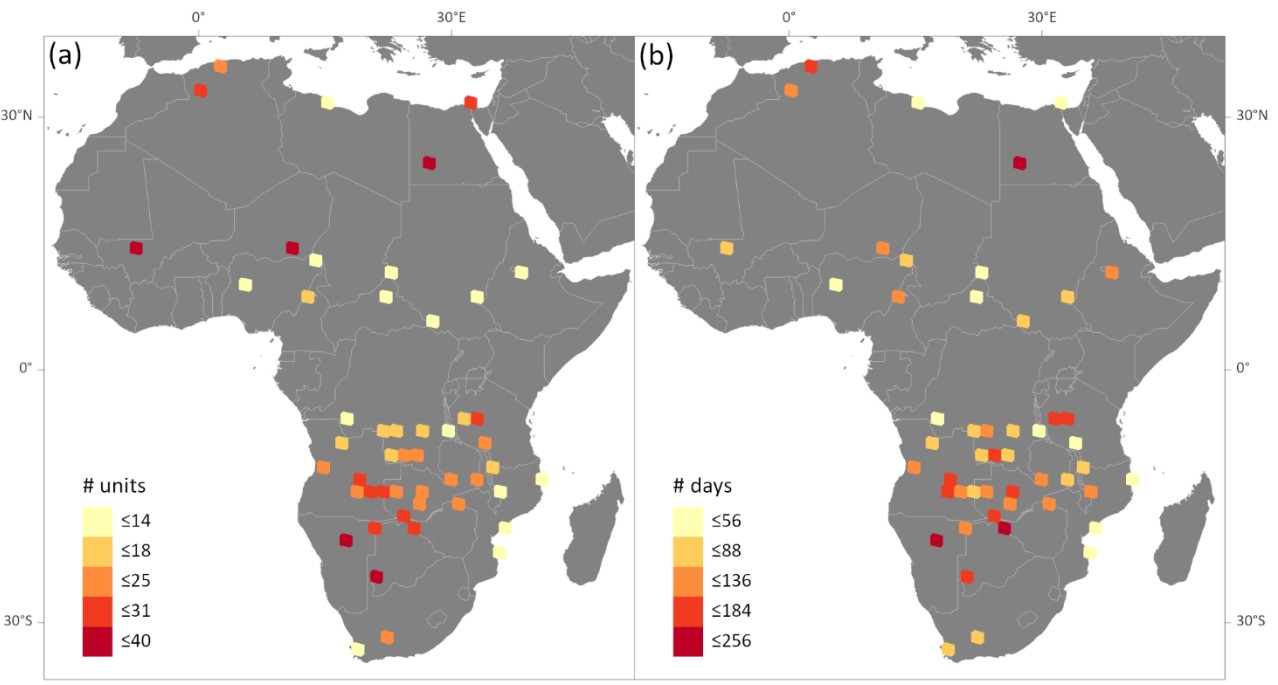

**Figure 6: Spatial distribution of the reference sites for the FireCCI Africa (2016) dataset: (a) number of short units interpreted in each validation site and (b) temporal length of the long units.**

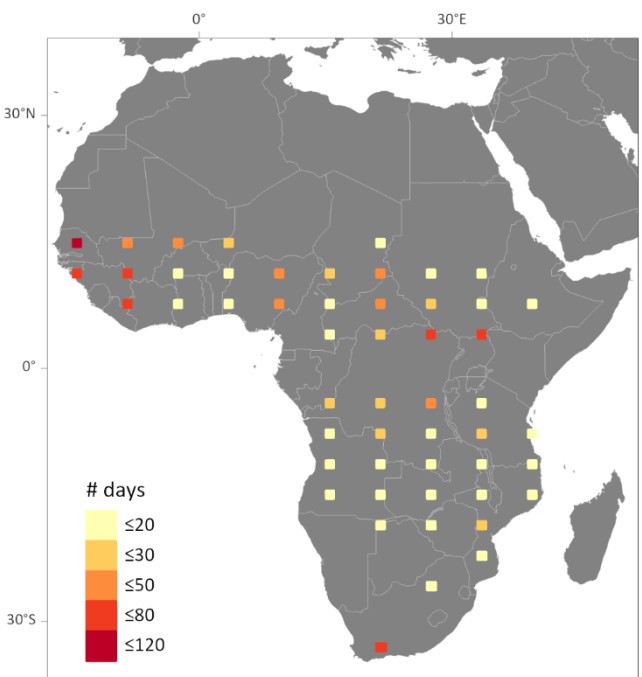

**Figure 7: Spatial distribution of the reference sites for FireCCI Africa S2 (2016) dataset. The legend shows the temporal distance (days) between the pre- and post-fire images used in each validation site for the year 2016.**

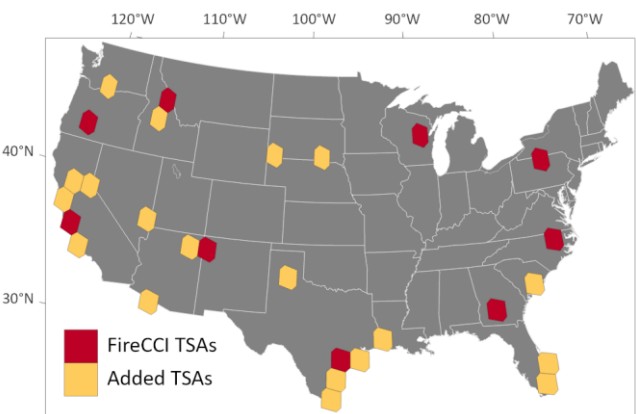

**Figure 8: Spatial distribution of the validation Thiessen scene areas (TSAs) for CONUS Landsat Burned Area (1988-2013) dataset. Modified from Vanderhoof et al. (2017). Reference data were generated for each TSA in each of the six sample years (1988, 1993, 1998, 2003, 2008, 2013).**

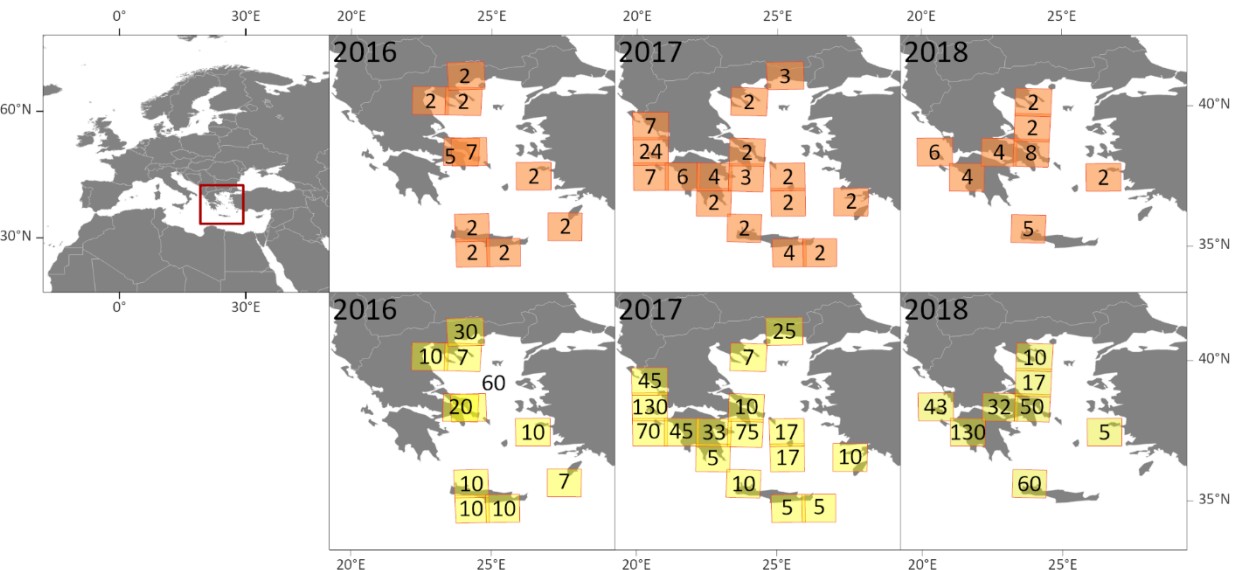

**Figure 9: Spatial distribution of validation sites for NOFFi Greece (2016-2018) reference dataset based on Sentinel-2 tiles. The orange figures above show the number of images used in each validation site for each year, whereas the yellow ones below show the temporal length (days) of the reference data files generated in each validation site.**


**Table 1: Example of the standard attribute table of the reference shapefiles.**

| category | preDate | postDate | preImg | postImg | path | row | year | area |
|---|---|---|---|---|---|---|---|---|
| 3 | 1988-07-05 | 1988-10-25 | LT50150351988187XXX05 | LT50150351988299XXX08 | 15 | 35 | 1988 | 267043.6 |
| 2 | 1988-07-05 | 1988-10-25 | LT50150351988187XXX05 | LT50150351988299XXX08 | 15 | 35 | 1988 | 4557.8 |
| 1 | 1988-07-05 | 1988-10-25 | LT50150351988187XXX05 | LT50150351988299XXX08 | 15 | 35 | 1988 | 2043.3 |
| 1 | 1988-07-05 | 1988-10-25 | LT50150351988187XXX05 | LT50150351988299XXX08 | 15 | 35 | 1988 | 900.4 |



**Table 2: Datasets included in the Burned Area Reference Database. CCI: Climate Change Initiative, CONUS: contiguous United States, NOFFi: National Observatory of Forest Fires, TM: Thematic Mapper, ETM+: Enhanced TM, OLI: Operational Land Imager, CEOS-LPVS: Committee on Earth Observing Satellites-Land Product Validation Subgroup, SRS: Stratified Random Sampling, SS: Systematic Sampling, NPS: Non-probability sampling.**

| Dataset | Project | Years | Extent | Source Imagery | Sampling Method | CEOS-LPVS Stage | Reference |
|---|---|---|---|---|---|---|---|
| FireCCI global (2008) | FireCCI | 2008 | global | Landsat TM, ETM+ | SRS | 3 | Padilla et al. (2014) |
| FireCCI global (2003-2014) | FireCCI | 2003-2014 | global | Landsat TM, ETM+, OLI | SRS | 3 | Padilla et al. (2018) |
| FireCCI Africa (2016) | FireCCI | 2016 | Africa | Landsat ETM+, OLI | SRS | 3 | Padilla et al. (2018) |
| FireCCI Africa S2 (2016) | FireCCI | 2016 | Africa | Sentinel-2 MSI | SS | 1 | Unpublished |
| CONUS Landsat Burned Area (1988-2013) | Landsat Level-3 Science Products | 1988, 1993, 1998, 2003, 2008, 2013 | United States | Landsat TM, ETM+, OLI | SRS | 4 | Vanderhoof et al. (2017;2020) |
| NOFFi Greece (2016-2018) | NOFFi | 2016-2018 | Greece | Sentinel-2 MSI | NPS | 1 | Unpublished |

**Table 3: Summary of the total area (km²) of the 3 mapped categories (burned, unburned and no-data) and percentage of each category respect the total area mapped for each dataset. Additionally, the total land surface and percentage respect the total area interpreted is provided. The region extent and the total number of reference files included in each dataset is also indicated.**

| Dataset | Region extent | Reference Files (#) | Burned (km²) | Unburned (km²) | No-data (km²) | Land surface (km²) | Total area (km²) |
|---|---|---|---|---|---|---|---|
| FireCCI global (2008) | L5: complete scene L7: central regions without SLC-off gaps | 105 | 23802.26 (1.35%) | 1558931.69 (88.35%) | 181761.84 (10.30%) | 1679627.66 (95.19%) | 1764495.79 |
| FireCCI global (2003-2014) | 30 x 20 km | 1200 | 27692.96 (3.85%) | 516396.61 (71.85%) | 174591.03 (24.29%) | 674926.47 (93.91%) | 718680.59 |
| FireCCI Africa (2016) SU | 30 x 20 km | 1052 | 8398.07 (1.33%) | 474349.56 (75.23%) | 147821.16 (23.44%) | 576181.91 (91.37%) | 630568.80 |
| FireCCI Africa (2016) LU | 30 x 20 km | 50 | 3663.84 (15.72%) | 11562.91 (49.61%) | 8081.50 (34.67%) | 20737.37 (88.97%) | 23308.25 |
| FireCCI Africa S2 (2016) | 110 x 110 km | 52 | 55583.10 (8.87%) | 454013.51 (72.42%) | 117317.47 (18.71%) | 616483.40 (98.34%) | 626914.08 |
| CONUS Landsat Burned Area (1988-2013) | L5-7-8: complete scene | 168 | 6226.45 (0.12%) | 4308711 (82.33%) | 918382.18 (17.55%) | 4251639.569 (81.24%) | 5233319.62 |
| NOFFi Greece (2016-2018) | 110 x 110 km | 34 | 398.62 (0.10%) | 105865.87 (25.83%) | 303640.87 (74.08%) | 129072.703 (31.49%) | 409905.36 |

**Table A1: FireCCI global (2008) stratified sampling data. Distribution of sampled ($n_h$) and total population ($N_h$) Thiessen scene areas (TSAs) by biome and BA stratum. BA: burned area.**

| Biome | Number of TSAs sampled ($n_h$) | | Total number of TSAs ($N_h$) | |
|---|---|---|---|---|
| | High BA stratum | Low BA stratum | High BA stratum | Low BA stratum |
| Boreal forest | 8 | 4 | 215 | 857 |
| Mediterranean forest | 4 | 3 | 28 | 113 |
| Others | 3 | 2 | 559 | 2148 |
| Temperate forest | 8 | 9 | 178 | 704 |
| Temperate grassland & savanna | 4 | 3 | 160 | 637 |
| Tropical forest | 9 | 7 | 174 | 696 |
| Tropical & Subtropical savanna | 12 | 29 | 151 | 602 |

**Table A2: FireCCI global (2003-2014) stratified sampling data. Distribution of sampled units ($n_h$) and total population ($N_h$) by year, biome and BA stratum. H: high, L: Low, BA: burned area.**

| Biome | 2003 | 2004 | 2005 | 2006 | 2007 | 2008 | 2009 | 2010 | 2011 | 2012 | 2013 | 2014 |
|---|---|---|---|---|---|---|---|---|---|---|---|---|
| **Boreal forest** | | | | | | | | | | | | |
| Sampled H BA | 2 | 2 | 2 | 2 | 2 | 2 | 2 | 2 | 2 | 2 | 2 | 2 |
| Sampled L BA | 2 | 2 | 2 | 2 | 2 | 2 | 2 | 2 | 2 | 2 | 2 | 2 |
| Population H BA | 752 | 745 | 1344 | 537 | 664 | 826 | 926 | 533 | 1295 | 1213 | 726 | 633 |
| Population L BA | 40924 | 47189 | 33173 | 33711 | 37976 | 35641 | 41324 | 37341 | 22503 | 26626 | 29644 | 35299 |
| **Mediterranean forest** | | | | | | | | | | | | |
| Sampled H BA | 2 | 2 | 2 | 2 | 2 | 2 | 2 | 2 | 2 | 2 | 2 | 2 |
| Sampled L BA | 2 | 2 | 2 | 2 | 2 | 2 | 2 | 2 | 2 | 2 | 2 | 2 |
| Population H BA | 179 | 287 | 212 | 292 | 217 | 346 | 329 | 269 | 247 | 314 | 223 | 172 |
| Population L BA | 8333 | 7116 | 7553 | 7139 | 7923 | 6853 | 7846 | 7202 | 7857 | 5516 | 7920 | 8789 |
| **Others** | | | | | | | | | | | | |
| Sampled H BA | 2 | 4 | 2 | 6 | 4 | 2 | 2 | 3 | 13 | 2 | 2 | 4 |
| Sampled L BA | 2 | 2 | 2 | 2 | 2 | 2 | 2 | 2 | 2 | 2 | 2 | 2 |
| Population H BA | 1694 | 791 | 996 | 768 | 734 | 494 | 798 | 792 | 1134 | 1043 | 709 | 764 |
| Population L BA | 68577 | 58049 | 58971 | 61564 | 59484 | 58978 | 62512 | 60303 | 55806 | 40999 | 60530 | 69961 |
| **Temperate forest** | | | | | | | | | | | | |
| Sampled H BA | 2 | 2 | 2 | 2 | 2 | 2 | 2 | 2 | 2 | 2 | 2 | 2 |
| Sampled L BA | 2 | 2 | 2 | 2 | 2 | 2 | 2 | 2 | 2 | 2 | 2 | 2 |
| Population H BA | 584 | 1343 | 1309 | 323 | 951 | 601 | 818 | 1021 | 907 | 345 | 748 | 729 |
| Population L BA | 38622 | 32424 | 32747 | 34122 | 33850 | 31544 | 34438 | 32708 | 33925 | 23146 | 29994 | 33036 |
| **Temperate grassland & savanna** | | | | | | | | | | | | |
| Sampled H BA | 5 | 3 | 4 | 4 | 4 | 6 | 5 | 3 | 3 | 3 | 3 | 5 |
| Sampled L BA | 2 | 2 | 2 | 2 | 2 | 2 | 2 | 2 | 2 | 2 | 2 | 2 |
| Population H BA | 1642 | 943 | 1220 | 996 | 985 | 1257 | 587 | 858 | 568 | 601 | 488 | 973 |
| Population L BA | 26124 | 24516 | 24402 | 24702 | 24697 | 23761 | 26517 | 25079 | 24804 | 17071 | 23684 | 25603 |
| **Tropical forest** | | | | | | | | | | | | |

| | | | | | | | | | | | | |
|---|---|---|---|---|---|---|---|---|---|---|---|---|
| Sampled H BA | 5 | 5 | 5 | 4 | 5 | 3 | 4 | 6 | 3 | 4 | 4 | 4 |
| Sampled L BA | 2 | 2 | 2 | 2 | 2 | 2 | 2 | 2 | 2 | 2 | 2 | 2 |
| Population H BA | 2433 | 1909 | 2052 | 1825 | 1701 | 1272 | 1731 | 1642 | 1548 | 1435 | 1210 | 1231 |
| Population L BA | 43609 | 42228 | 42188 | 40038 | 41325 | 41673 | 41109 | 41137 | 40775 | 27552 | 38253 | 40208 |
| **Tropical & subtropical savanna** | | | | | | | | | | | | |
| Sampled H BA | 61 | 62 | 55 | 50 | 55 | 60 | 61 | 60 | 50 | 64 | 55 | 62 |
| Sampled L BA | 9 | 8 | 16 | 18 | 14 | 11 | 10 | 10 | 13 | 9 | 10 | 7 |
| Population H BA | 4662 | 4673 | 2974 | 2153 | 3559 | 3646 | 3727 | 4660 | 3119 | 3195 | 3496 | 3918 |
| Population L BA | 22878 | 22496 | 24916 | 25124 | 23098 | 23049 | 22997 | 22343 | 22503 | 15632 | 23228 | 26382 |

**Table A3: FireCCI Africa (2016) stratified sampling data. Distribution of sampled long units and total population by biome and stratum. BA: burned area.**

| Biome | Number of sampled units ($n_h$) | | Total number of units ($N_h$) | |
|---|---|---|---|---|
| | High BA stratum | Low BA stratum | High BA stratum | Low BA stratum |
| Mediterranean forest | 2 | 2 | 22 | 120 |
| Others | 2 | 2 | 20 | 549 |
| Temperate grassland & savanna | 2 | 2 | 24 | 82 |
| Tropical forest | 2 | 2 | 96 | 220 |
| Tropical & subtropical savanna | 32 | 2 | 393 | 709 |

**Table A4: CONUS Landsat Burned Area (1988-2013) stratified sampling data. Distribution of sampled and population Thiessen scene areas (TSAs) by biome and stratum. Each sampled TSA was then sampled for 5 separate years; however, high/low BA stratum**
**was determined from 2008, alone. Total number of TSAs is calculated for the contiguous United States (CONUS). BA: burned area.**

| Biome | Number of TSAs sampled ($n_h$) | | Total number of TSAs ($N_h$) | |
|---|---|---|---|---|
| | High BA stratum | Low BA stratum | High BA stratum | Low BA stratum |
| Temperate forest | 6 | 5 | 45 | 179 |
| Mediterranean forest | 2 | 1 | 2 | 10 |
| Temperate grassland & savanna | 2 | 3 | 25 | 99 |
| Tropical & subtropical savanna | 2 | 2 | 2 | 5 |
| Xeric/desert shrub | 3 | 2 | 17 | 66 |

