# Peer review of "Development of a standard database of reference sites for validating global burned area products"

_Earth System Science Data, 2020_

## Referee Comment (RC1) · Anonymous Referee #1 · 21 Jun 2020

General

The reference datasets for validating global burned area products provide a valuable resource to the fire mapping community. As the authors note, collecting reference data to validate burned area products is an expensive and time consuming proposition. Having available a vetted set of reference sample sites for map producers to readily access will greatly enhance the quantity and quality of information available to assess and compare accuracy of burned area products. The global extent of these datasets will facilitate regional comparisons as well, as users of the data will be able to extract data specific to their study area. One of the fundamental challenges of mapping of any

theme, burned area or otherwise, is the immense difficulty of obtaining reference data. The burned area reference dataset (BARD) presented by the authors is a significant advance to diminish this difficulty.

Specific Comments

1. The authors astutely identify the role of sampling in the collection of these burned area reference datasets (Line 63). It would be useful to add some explanation distinguishing between reference data collected by a formal sampling design, often called probability sampling designs, and reference data collected by convenience, ease of access, or other method that does not necessarily have randomization. Reference data collected by a randomized sampling design are suitable to support rigorous statistical statements about accuracy, whereas data collected by convenience can be suspect in this regard (i.e., data may not be representative of the entire area of interest). The implications of how the reference data were obtained should be noted. The manuscript clearly indicates that the Boschetti et al. (2019) and Padilla et al. (2014:2015) reference datasets were obtained from locations selected by stratified random sampling. For some of the other datasets, this is less clear. It would be useful for the authors to check each dataset and be sure that it is indicated whether the dataset had an underlying randomized sampling design.

2. Related to the previous comment, the manuscript identifies that several of the datasets included were selected by stratified sampling designs, and these designs had intensified sampling in high burned area strata. According to the original articles associated with these datasets, rather complex estimation formulas have to be applied to such data (i.e., the less complicated formulas of simple random sampling are not appropriate when the sampling was stratified with different sampling intensities in the strata). It would therefore seem necessary that users of these reference datasets be cautioned about the need to use proper estimation formulas if users are to correctly report accuracy from these stratified sample datasets. This would also create the need to include in the datasets the information required to apply these estimation formulas,

for example stratum sizes, the stratum ID of each sampled unit, and perhaps additional information depending on the specific details of the particular dataset.

Technical Corrections and Suggestions

Throughout, readability would be enhanced by using paragraph indents at the start of each paragraph.

L23: insert "a" to revise to "requires a high level"

L26, L29: Given that the acronym BARD was defined at Line 26, replace "The Database" with "BARD"

L40: "sensors" should be "sensor"

L41: revise to "reference data that are based on" ["data" is plural so "data that are"]

L46: "products" instead of "product efforts"

L63, L79, L105, L106, L159, L161, L164, L198, L205, L207, L209: Throughout the manuscript, the words "file" and "files" are sometimes used to the refer to the actual reference data. For example at L63, the "files" were not derived from pairs of images, but rather the "reference data" that are stored in the files have been produced from the pairs of images. The text should be revised to replace "files" with "reference data" unless the text is referring to the actual files that store the reference data.

L64: Replace "without probabilistic meaning" by "that were not selected using a probability sampling design". It is not clear what "direct sampling" is. Is direct sampling convenience, purposeful, or other sampling without randomization?

For all examples at Lines 65-70, it appears that there was a rationale for why sites were selected (even if they were not selected by a randomized protocol). It would be useful to mention what purposeful selection criteria were used. The Roy and Boschetti example mentions sites selected to be spatially distributed across the landscape, so this is an example where the manuscript provides useful additional information regarding the

purposeful selection criteria.

L70-71: If Boschetti et al. (2019) collected data for only a single year, does that qualify as a "full spatio-temporal validation"? It would be helpful to define what a "full validation" is in regard to time and space.

L88: insert "design" after "random sampling" to create "stratified random sampling design"

L89: Consider revising to: "Boschetti et al. (2016) extended the sampling design to include the temporal dimension of the sampling units."

L90: insert "the" between "allocate sample" and delete "a" from "example a stratified"

L91: insert "the" before "sample"

L94: replace "are" by "is" because "dimension" is a singular noun

L99: delete "a"

L106: Consider revising to: "The procedures implemented to obtain those burn patches are diverse, depending . . ."

L109-110: Consider revising to: "Parts of the scene that cannot be observed or interpreted because of clouds or sensor problems (i.e., Scan Line . . ."

L115: replace "such" by "each" and replace "like" by "such as"

L153: Are n=127 and n=131 the number of TSAs sampled? It is not clear what these numbers represent.

L170: delete "to each sample unit" because this threshold is applied to all TSAs. That is, all TSAs are assigned to strata as part of the sample selection process. It is not just the sampled units that are assigned to strata.

L172: given that "proportional allocation" for stratified sampling is defined as the sample size in each stratum being proportional to the number of units in the entire study

region belonging to that stratum, replace "applying a proportional allocation" by "applying a sample allocation".

L182: replace "in" with "of" and replace "days" with "day"

L185: It is not clear how the actual time period covered by these "long units" is defined.

L186: Consider revising to: "Reference maps using long units concatenate consecutive 8-16 day maps (Fig. 5)."

L188: The 50 units are for fire CCI Africa compared to 100 units per year for FireCCI global?

L189: replace "consists on" with "consists of" and replace "perimeters" by "perimeter"

L190: replace "units" by "unit" (2 cases) and "days" by "day"

L198: remove "A" before "systematic sampling"

L201: replace "the whole" with "all" and replace "was" with "were"

L203: "consecutively" should be "sequentially"

L209: "joined" should be "joint" and "by" should be "between"

L213: delete "the" before "77%"

L219: replace "scar samples" by "scars sampled"

L223: "days" should be "day"

L224: "pair" should be "pairs"

L228-229: Continue to use the same phrasing as at L180 and L207 to identify the stage of the reference dataset. The sentence structure at L180 and L207 is much easier to read.

L231-232: replace "generated to perform the validation of the BAECV" with "generate

to validate the BAECV"

L232: Move the text "Landsat Burned Area Essential Climate Variable" to before the first use of BAECV at Line 231.

L238: delete "A" before "systematic"

L239: the three values of n sum to 335 images not 336

L243: replace "... only two (pre and post-fire image..." by "...only two images (pre- and post-fire)..."

L266-267: Continue to use the same phrasing as at L180 and L207 to identify the stage of the reference dataset.

L272: "wildfires" should be "wildfire"

L279: "were" should be "was"

L283: "postfire" should be "post-fire"

L284: "formers" should be "former"

L290-291: Continue to use the same phrasing as at L180 and L207 to identify the stage of the reference dataset.

L306: Consider changing "futures updates come to replace the lack..." with "future updates remedy the lack..."

---

## Short Comment (SC1) · 28 Jul 2020

Table 2 summarizes the number of reference files but does not provide summary information on the total areas of the 3 mapped classes (burned, unburned, no data).

Please add a new table providing this information for each project and for all the projects together. This is needed because (i) Landsat and Sentinel-2 images have different areas (∼185x185 km and ∼110x110 km), (ii) different image spatial subsets were mapped (i.e., not the entire image extents) by the different projects, (iii) the "no data" class includes areas where the interpreter did not undertake the mapping and it is unclear if this was a major proportion and/or if it varied among the projects.

It would be helpful to then place the total mapped burned and unburned areas in the context of (a) the total global land area and (b) the typical total annual global area burned, and update the abstract and conclusion accordingly.

---

## Short Comment (SC2) · 28 Jul 2020

Could you explain the long and short units methodology better please. Currently as written I cannot understand it. Figure 5 is helpful but it does not show the case where there are "no data" areas (for example, due to cloud and/or the Landsat SLC-off issue) in the image time series. Please clarify this in the paper text - paying particular attention to how the "no data" pixels are handled in the long unit derivation. I suspect that there are underlying assumptions that reduce the utility of the long unit results for validation. For example, it is well known that in many regions the burn signal dissipates rapidly and that clouds occur commonly and often at the time of Landsat or Sentinel-2 overpass.

Thus, to my mind, the long unit may (i) fail to capture the true area burned over the time series, (ii) reduce the proportion of the image area that is mapped as burned and/or unburned. Please clarify and discuss.

———————————————————————

---

## Short Comment (SC3) · 28 Jul 2020

Given the effort and cost involved in generating validation dataset, the standardization and documentation of existing datasets for future use is certainly a meritorious effort, and there is no doubt that the datasets made available by the authors of this paper will find a use in the fire community.

I have however some concerns.

1) Scope of the paper and qualification of the dataset The way the dataset is presented could lead to some confusion and misinterpretation. The paper title refers to the 'de-

velopment of a standard database' and the abstract refers to the present work as the compilation of 'the first Burned Area Reference Database'. This is misleading, because the work described in the present paper is limited to the collation of existing datasets, through standard GIS operations described in Section 2.3, namely the conversion of the various datasets to the same file format (shapefile), the use of standardized file names and the creation of simple metadata (Table 1). The wording of the abstract, i.e. referring to BARD as 'the first Burned Area Reference Database', is incorrect, as this is not the first burned area reference dataset - all the datasets that constitute the BARD are pre-existing. Maybe 'the first publicly available burned area reference dataset' would be a more appropriate statement.

2) Degree of novelty Section 2.3 is the only section that reports original work (i.e. the conversion of all data to shapefile, the standard filenames and the metadata), while the rest of the methods document what was done by the various research teams in the projects that provided the data.

3) Sampling. Section 2.1 ('Selection of the validation sites') describes a procedure for stratified random sampling of the burned area reference data that was followed by some of the collated datasets (but not all), which is extremely confusing. An inattentive reader might be led to believe that the BARD dataset itself is the result of a stratified random sampling, rather than the collation of datasets some of which were the results of stratified random sampling (albeit with different methods) and some that are not.

4) Stage 3 validation data set. Much is said throughout the paper of the compliance with CEOS Stage 3 validation requirements, but the BARD dataset as currently defined does not meet those requirements, i.e. it would not allow for the use of unbiased estimators of the accuracy metrics, and their associated standard errors. In the current form, pieces of BARD could be used for a Stage 3 validation, whereas other pieces could only be used for a Stage 1 or Stage 2 validation. Could the authors provide a harmonized statistical framework for the estimation of accuracy metrics from the whole BARD dataset?

5) Burned area minimum mapping unit. Was the minimum mapping unit the same among the projects? This is a particularly relevant issue, as the omission of small burned areas is one of the primary areas of interest for the validation of coarse resolution burned area products.

6) Quality assessment The 'methods' do not include any section on quality assessment/assurance yet the abstract states that 'All reference files have been checked for internal quality'. In particular, there is no corresponding methods section detailing how the quality was assessed – for example, were the reference burned area maps opened by an interpreter, and checked to ensure that there are no obvious omission or commission errors either in the burned area perimeters, or in the cloud and shadow masks that constitute the 'no data' class ? Were all the data of the same quality ?

---

## Short Comment (SC4) · 28 Jul 2020

Dataset harmonization It is apparent that the individual datasets collated in BARD were derived using a variety of semi-automatic procedures, and in the context of projects that had a somewhat different emphasis. How were the data harmonized so that they can be used together meaningfully ? The title, abstract, introduction and conclusion imply a degree of harmonization between the datasets that is well beyond what was done, and might be seen as overstating the potential of the BARD to be a 'standard database'. Furthermore, there is no formal discussion in the paper of what requirements/criteria/standards should be met by a 'standard database'.

---

## Referee Comment (RC2) · Anonymous Referee #2 · 4 Aug 2020

General Comments

This manuscript describes the availability of a new dataset comprising a compilation of reference burned area data, which can be used for the validation of burned area products. The short description paper outlines the methods used to standardise a number of different datasets into a common format, and a more detailed description on each one. It also gives an overview of why validation is necessary but not always readily available, which provides useful context.

Validation of burned area products is definitely lacking in the field, and this is a welcome contribution to the research area. I think it will be useful for many researchers working

on fire and burned area. The methods are well-referenced, and are mostly clearly described, with the exception of a few points as outlined below.

The data is readily available via the link provided in the text, and can be accessed immediately after completing a short form. The data appears complete.

Specific comments

Section 2.1 Selection of validation sites: This section comes across as a literature review of different methods, and I'm not sure what is actually being implemented in this paper from reading this section. Can you clarify in the paragraph (e.g. does each dataset use a different method?)

At the end of the Introduction, the overview of the paper is a bit vague. I think this would benefit from a clearer outline of the structure, and a list of the datasets that are considered in this paper to give a better overview up front.

Line 220 – only data in June to October is considered for this dataset. This covers the main fire season in this region, but how are the fires outside of the fire season dealt with?

Presumably the temporal length of the reference files is such that it covers multi-day burning. It is worth pointing this out in the text explicitly.

It would be useful to include some text describing how one might use all these different reference datasets in practise. Should they all be used together, and if so how should the range be accounted for?

How were these datasets selected? Are there any other datasets available that are not included here, or are these the only ones available? I suggest including some explanation of this in the text.

Are all the datasets related to FireCCI? It seems so from the description of the data via the link, but not in the paper.
Most modellers use NetCDF, if it would be nice if this format was considered for future releases.

Technical corrections

References to figures ("Fig.") throughout the text is sometimes with a space and sometimes without

Line 49 – change to "acquired in the year 2000"

Line 182 - "consists of"

---

## Short Comment (SC5) · 4 Aug 2020

The standard datasets are relevant for validation of burned areas and the scientific community will be interested in such product. However, I found quality issues that limit the application without additional evaluation of files. Since these datasets have certain degree of automation in the production, further inspection is required to guarantee the high-quality in reference files. For instance, some areas present no-data/unobserved labels without a clear reason. When the reference dataset omits complex burned areas, the validation results tend to be biased. Other problems were observed in water pixels labeled as "unburned", and harvest areas as "burned". The authors should ac-

[Figure]

knowledge the problems in these reference files and improve the quality as much as possible.

[Figure]

**Fig. 1.**

[Figure]

---

## Short Comment (SC6) · 5 Aug 2020

I examined all the Brazilian (BrFLAS) data including comparing them to the multi-date Landsat images they were derived from. Two obvious issues:

1) None of the Brazilian data have a "no data\unobserved" class. This would only be correct if the images were always cloud- and shadow- free and but this is not the case. For example, see below.

2) There are burned areas that are not mapped as "burned" because one of the images was cloud/shadow obscured. However, incorrectly, they have not been mapped as "no

[Figure]

data\unobserved" (for example, see in red circle below). This makes these data difficult to use for validation, or as a reliable source of training data for classification purposes (as without looking at the images I would assume incorrectly that these areas were unburned).
* * *
[Figure]

[Figure]

LC08_L1TP_219063_20150923    LC08_L1TP_219063_20151025

[Figure]

BrFLAS_RD_219063_20150923_20151025

**Fig. 1.**

---

## Referee Comment (RC3) · Anonymous Referee #3 · 27 Aug 2020

**General Comments**

This manuscript describes a first attempt at compiling a common database of burned area reference perimeters ("BARD") suitable for validating remotely sensed burned area data sets. While the goal of producing the BARD is laudable, I feel the result falls somewhat short in that the authors provide no guidance in how this dataset should be used in practice. While reprojected and vectorized into a common format, the six underlying validation data sets were not generated in an entirely consistent manner and vary significantly in terms of sampling strategy and minimum mapping unit and

various other important respects. As a result, I feel it is essential for the authors to advise users how the database as a whole should be used. For example, should some or all of the individual data sets be merged, or should they always be used separately? If the latter, then any validation of a global data set will yield six different sets of validation results. How should these results be interpreted, especially for the individual data sets that overlap in space and time, such as FireCCI Africa and FireCCI Africa S2? Furthermore, the authors state that "The Burned Area Reference Database will be expanded with new reference files that are being produced in the FireCCI project and we encourage future contributions from the scientific community", but it is not clear how this plan can scale practically as the number of data sets grows.

**Specific Comments**

L41: "they require generating global reference data that is based on higher-resolution sensors" Although I agree with this statement, it overlooks studies such as Roteta et al. (2019) who used 30-m Landsat images to validate a 20-m Sentinel-2 burned area product.

L59: Giglio et al. (2018) give a release date of mid 2008 for the NASA MCD45A1 product.

L68: "The MCD64A1 Collection 5 was not formally validated" Giglio et al. (2009) performed an "accuracy assessment for three geographically diverse regions (central Siberia, the western United States, and southern Africa)" using 50 Landsat scenes. Is this not validation?

L85/Section 2.1: The authors note the importance of sampling design and describe
various important components of this process, but not all of the BARD data sets seem to have adopted the strategies described in this section. It would be helpful to note any deviations within BARD from the sampling strategy described here. The authors might perhaps also provide a brief summary of the CEOS-LPVS validation stages to help readers interpret the stage numbers mentioned later for the individual data sets (in Table 2, for example) in the context of sampling.

L158: "The FireCCI global 2008 dataset includes 129 reference data files" This number differs slightly from Padilla et al. (2014), who refer to "102 sampled pairs". Presumably additional scenes were added to that data set. This is worth mentioning since it would alert readers that the summaries and/or statistics provided in Padilla et al. (2014) do not necessarily apply to the FireCCI global (2008) distributed in BARD.

L195/Section 2.4.4: The 2016 FireCCI Africa S2 data set is not mentioned in either of the references cited in this section. Please add the correct reference or clarify that the data set has not been previously published.

L208/Section 2.4.5: Rodrigues et al. (2019) mention a minimum mapping unit of 21 ha. Does this threshold also apply to the BrFLAS Brazil data distributed in the BARD?

L230/Section 2.4.6: Hawbaker et al. (2020) include the following remark about the BAECV validation data set: "Because no independent reference data were available for burned areas in agricultural cover types, the Landsat-based BAMS reference dataset did not train on agricultural fires and consequently cannot be considered accurate for this cover type." Have the unreliable reference polygons belonging to this category been flagged or removed from BARD? If not, some guidance to users about how they should identify and handle such cases would be appropriate.

L242: "The pre- and post-fire image pairs did not specifically represent a probability sample within a year but were designed to target changes incurred over the peak fire season." Given this targeting of the peak fire season, is it appropriate to use this data set for assessing out of season commission errors?

L268/Section 2.4.7: Given that the NOFFi-OBAM mapping service "is activated after large wildfires events and under explicit requests by the local forest offices", is it appropriate to use this data set for assessing commission errors? Please explain and include appropriate caveats if necessary.

Figure 3: This figure shows perhaps a dozen validation sites that are not shown in the equivalent figure of Padilla et al. (2014), where the 2008 FireCCI global validation data set was originally described. Please see related L158 comment above.

Figure 5 would be much more useful if it included clouds or some other source of missing data in the Landsat image stack. The long unit sampling is not clearly described in the manuscript, but I think I understand most of what the authors here poorly describe only after consulting Figure 12 of Padilla et al. (2018). Perhaps the authors could include a similar figure here.

Figure 9: Not clear why it is useful to highlight FireCCI TSAs vs. Added TSAs on the map. It would be more useful and more consistent to show the time period between Landsat image pairs as was done for the other data sets in Figures 3, 6, 7, and 8.

Table 2: Please show the total areas of the separate burned, unburned, and no-data

[Figure]

classes for each data set.

BARD DOI landing page (https://edatos.consorciomadrono.es/dataset.xhtml?persistentId=doi:10.21950/BBQQU7): The landing page describes BARD almost exclusively as a FireCCI effort. This is a little bit inconsistent with the manuscript, which says that the database "was created by compiling existing reference burned area datasets from different international projects."

**Technical Corrections**

L40: change "sensors" to "sensor"

L57-58: Acronyms MERIS and MODIS not defined

L85: change "amount" to "number"

L91: change "sample" to "samples"

L213: change "covering the 77%" to "covering 77%"

Figure 2 caption: change "Time distance between" to "Time period between"

———————————————

---

## Author Comment (AC7) · 2 Sep 2020

General

The reference datasets for validating global burned area products provide a valuable resource to the fire mapping community. As the authors note, collecting reference data to validate burned area products is an expensive and time consuming proposition. Having available a vetted set of reference sample sites for map producers to readily access will greatly enhance the quantity and quality of information available to assess and compare accuracy of burned area products. The global extent of these datasets will facilitate regional comparisons as well, as users of the data will be able to extract data specific to their study area. One of the fundamental challenges of mapping of any theme, burned area or otherwise, is the immense difficulty of obtaining reference data. The burned area reference dataset (BARD) presented by the authors is a significant advance to diminish this difficulty.

We sincerely appreciate this review and thank the positive comments about the contribution of this manuscript to the field.

Specific Comments

1. The authors astutely identify the role of sampling in the collection of these burned area reference datasets (Line 63). It would be useful to add some explanation distinguishing between reference data collected by a formal sampling design, often called probability sampling designs, and reference data collected by convenience, ease of access, or other method that does not necessarily have randomization. Reference data collected by a randomized sampling design are suitable to support rigorous statistical statements about accuracy, whereas data collected by convenience can be suspect in this regard (i.e., data may not be representative of the entire area of interest). The implications of how the reference data were obtained should be noted. The manuscript clearly indicates that the Boschetti et al. (2019) and Padilla et al. (2014:2015) reference datasets were obtained from locations selected by stratified random sampling. For some of the other datasets, this is less clear. It would be useful for the authors to check each dataset and be sure that it is indicated whether the dataset had an underlying randomized sampling design.

Response: This is indeed a very useful add in to the description. We have extended section 2.1 to better explain these aspects. We have revised the detailed description of each dataset and have included which sampling method was used. Please, also note that the sampling design used in each dataset has been summarized in table 2.

2. Related to the previous comment, the manuscript identifies that several of the datasets included were selected by stratified sampling designs, and these designs had intensified sampling in high burned area strata. According to the original articles associated with these datasets, rather complex estimation formulas have to be applied to such data (i.e., the less complicated formulas of simple random sampling are not appropriate when the sampling was stratified with different sampling intensities in the strata). It would therefore seem necessary that users of these reference datasets be cautioned about the need to use proper estimation formulas if users are to correctly report accuracy from these stratified sample datasets. This would also create the need to include in the datasets the information required to apply these estimation formulas, for example stratum sizes, the stratum ID of each sampled unit, and perhaps additional information depending on the specific details of the particular dataset.

Response: Thank you for this observation. We have added the required data to use the validation datasets obtained through SRS to make probabilistic estimations of accuracy. The stratum ID of each sampled unit and the total area of the TSAs from which reference data was obtained have been added to the .csv files provided in the metadata folder. In addition, a table with the stratum sizes for each reference dataset is also provided in section 5 (Appendix A: Supplementary tables).

Technical Corrections and Suggestions

Throughout, readability would be enhanced by using paragraph indents at the start of each paragraph.

Response: Preprint manuscript was prepared according to the journal template, post-editing will be applied to the final version.

L23: insert "a" to revise to "requires a high level"

Response: Done.

L26, L29: Given that the acronym BARD was defined at Line 26, replace "The Database" with "BARD"

Response: Done.

L40: "sensors" should be "sensor"

Response: Done.

L41: revise to "reference data that are based on" ["data" is plural so "data that are"]

Response: Done.

L46: "products" instead of "product efforts"

Response: Done.

L63, L79, L105, L106, L159, L161, L164, L198, L205, L207, L209: Throughout the manuscript, the words "file" and "files" are sometimes used to the refer to the actual reference data. For example, at L63, the "files" were not derived from pairs of images, but rather the "reference data" that are stored in the files have been produced from the pairs of images. The text should be revised to replace "files" with "reference data" unless the text is referring to the actual files that store the reference data.

Response: Thank you for this observation, we have revised the document and changed it as suggested.

L64: Replace "without probabilistic meaning" by "that were not selected using a probability sampling design". It is not clear what "direct sampling" is. Is direct sampling convenience, purposeful, or other sampling without randomization?

Response: We have modified the sentence to '*Early validation exercises were subjected to a first stage validation, usually based on small samples of reference sites that were not selected using a probability sampling design, but rather by a purposeful or convenience selection based on data availability or expert knowledge to ensure diverse wildfire conditions were included in the sample*'.

For all examples at Lines 65-70, it appears that there was a rationale for why sites were selected (even if they were not selected by a randomized protocol). It would be useful to mention what purposeful selection criteria were used. The Roy and Boschetti example mentions sites selected to be spatially distributed across the landscape, so this is an example where the manuscript provides useful additional information regarding the purposeful selection criteria.

Response: Validation sites selection in Chuvieco et al. (2008) was based on Landsat and CBERS images donated by regional space agencies, when Landsat archive wasn't free open to the public. We have mentioned it in the corresponding paragraph.

L70-71: If Boschetti et al. (2019) collected data for only a single year, does that qualify as a "full spatio-temporal validation"? It would be helpful to define what a "full validation" is in regard to time and space.

Response: We have removed the expression 'full spatio-temporal validation' to avoid confusion and changed the sentence to '*A recent study has provided a validation of the MCD64A1 product implementing a probability sampling design and using Landsat-8 Operational Land Imager (OLI) images but only for a single year (Boschetti et al., 2019)*'.

L88: insert "design" after "random sampling" to create "stratified random sampling design"

Response: Done.

L89: Consider revising to: "Boschetti et al. (2016) extended the sampling design to include the temporal dimension of the sampling units."

Response: Done.

L90: insert "the" between "allocate sample" and delete "a" from "example a stratified"

Response: Done.

L91: insert "the" before "sample"

Response: Done.

L94: replace "are" by "is" because "dimension" is a singular noun

Response: Done.

L99: delete "a"

Response: Done.

L106: Consider revising to: "The procedures implemented to obtain those burn patches are diverse, depending…"

Response: Done, text has been modified as suggested.

L109-110: Consider revising to: "Parts of the scene that cannot be observed or interpreted because of clouds or sensor problems (i.e., Scan Line …"

Response: Done, text has been modified as suggested.

L115: replace "such" by "each" and replace "like" by "such as"

Response: Done.

L153: Are n=127 and n=131 the number of TSAs sampled? It is not clear what these numbers represent.

Response: The numbers refer to the number of images interpreted from each sensor, 127 images from Landsat-5 and 131 from Landsat-7. We have changed the sentence to clarify this point. Please, note that numbers have been modified because we initially included some reference data that shouldn't be included in this dataset.

*'A total of 210 images from Landsat-5 TM (n=101) and Landsat-7 ETM+ (n=109) satellite sensors were used to retrieve BA perimeters'.*

L170: delete "to each sample unit" because this threshold is applied to all TSAs. That is, all TSAs are assigned to strata as part of the sample selection process. It is not just the sampled units that are assigned to strata.

Response: Thank you for pointing that out, we've changed the text according to your observation.

L172: given that "proportional allocation" for stratified sampling is defined as the sample size in each stratum being proportional to the number of units in the entire study region belonging to that stratum, replace "applying a proportional allocation" by "applying a sample allocation".

Response: Done.

L182: replace "in" with "of" and replace "days" with "day"

Response: Done.

L185: It is not clear how the actual time period covered by these "long units" is defined.

Response: The long sampling units are defined by multiple consecutive pairs of images (short sampling units, separated by 16 days or less) covering at least 100 days. We have clarified the concept of short and long units in section 2.2.

L186: Consider revising to: "Reference maps using long units concatenate consecutive 8-16 day maps (Fig. 5)."

Response: This line has been removed as long unit reference data generation methodology is now explained in section 2.2.

L188: The 50 units are for fire CCI Africa compared to 100 units per year for FireCCI global?

Response: The authors used different sampling intensities for Africa and global. The 50 (long) units for FireCCI Africa implied an effort in the generation of reference data similar to that for the 12 years of FireCCI global. In the former case, 1052 pairs of images were processed, and on the latter case 1200.

L189: replace "consists on" with "consists of" and replace "perimeters" by "perimeter"

Response: Done

L190: replace "units" by "unit" (2 cases) and "days" by "day"

Response: Done

L198: remove "A" before "systematic sampling"

Response: Done

L201: replace "the whole" with "all" and replace "was" with "were"

Response: Done

L203: "consecutively" should be "sequentially"

Response: Done

L209: "joined" should be "joint" and "by" should be "between"

Response: BrFLAS dataset has been removed from BARD (please, see SC6 response)

L213: delete "the" before "77%"

Response: BrFLAS dataset has been removed from BARD (please, see SC6 response)

L219: replace "scar samples" by "scars sampled"

Response: BrFLAS dataset has been removed from BARD (please, see SC6 response)

L223: "days" should be "day"

Response: BrFLAS dataset has been removed from BARD (please, see SC6 response)

L224: "pair" should be "pairs"

Response: BrFLAS dataset has been removed from BARD (please, see SC6 response)

L228-229: Continue to use the same phrasing as at L180 and L207 to identify the stage of the reference dataset. The sentence structure at L180 and L207 is much easier to read.

Response: BrFLAS dataset has been removed from BARD (please, see SC6 response)

L231-232: replace "generated to perform the validation of the BAECV" with "generate to validate the BAECV"

Response: As we have renamed BAECV dataset to CONUS Landsat Burned Area, the sentence "generated to perform the validation of the BAECV" has been replaced by "generate to validate the Landsat Burned Area product"

L232: Move the text "Landsat Burned Area Essential Climate Variable" to before the first use of BAECV at Line 231.

Response: As we have renamed BAECV dataset to CONUS Landsat Burned Area, the sentence has been changed to: '*The Landsat Burned Area reference dataset (Vanderhoof et al., 2017;2020) extends across the contiguous United States (CONUS) and was generate to validate the Landsat Burned Area product (Hawbaker et al., 2017;2020)*'.

L238: delete "A" before "systematic"

Response: Done

L239: the three values of n sum to 335 images not 336

Response: Thank you, the error has been corrected

L243: replace "…only two (pre and post-fire image…" by "…only two images (pre and post-fire) …"

Response: Done

L266-267: Continue to use the same phrasing as at L180 and L207 to identify the stage of the reference dataset.

Response: Done

L272: "wildfires" should be "wildfire"

Response: Done

L279: "were" should be "was"

Response: Done

L283: "postfire" should be "post-fire"

Response: Done

L284: "formers" should be "former"

Response: Done

L290-291: Continue to use the same phrasing as at L180 and L207 to identify the stage of the reference dataset.

Response: Done

L306: Consider changing "futures updates come to replace the lack…" with "future updates remedy the lack…"

Response: Done, text has been modified as suggested.

---

## Author Response (AR1)

###############################################################################

Anonymous Referee #1

General

The reference datasets for validating global burned area products provide a valuable resource to the fire mapping community. As the authors note, collecting reference data to validate burned area products is an expensive and time consuming proposition. Having available a vetted set of reference sample sites for map producers to readily access will greatly enhance the quantity and quality of information available to assess and compare accuracy of burned area products. The global extent of these datasets will facilitate regional comparisons as well, as users of the data will be able to extract data specific to their study area. One of the fundamental challenges of mapping of any theme, burned area or otherwise, is the immense difficulty of obtaining reference data. The burned area reference dataset (BARD) presented by the authors is a significant advance to diminish this difficulty.

We sincerely appreciate this review and thank the positive comments about the contribution of this manuscript to the field.

Specific Comments

1. The authors astutely identify the role of sampling in the collection of these burned area reference datasets (Line 63). It would be useful to add some explanation distinguishing between reference data collected by a formal sampling design, often called probability sampling designs, and reference data collected by convenience, ease of access, or other method that does not necessarily have randomization. Reference data collected by a randomized sampling design are suitable to support rigorous statistical statements about accuracy, whereas data collected by convenience can be suspect in this regard (i.e., data may not be representative of the entire area of interest). The implications of how the reference data were obtained should be noted. The manuscript clearly indicates that the Boschetti et al. (2019) and Padilla et al. (2014:2015) reference datasets were obtained from locations selected by stratified random sampling. For some of the other datasets, this is less clear. It would be useful for the authors to check each dataset and be sure that it is indicated whether the dataset had an underlying randomized sampling design.

This is indeed a very useful add in to the description. We have extended section 2.1 to better explain these aspects. We have revised the detailed description of each dataset and have included which sampling method was used. Please, also note that the sampling design used in each dataset has been summarized in table 2.

2. Related to the previous comment, the manuscript identifies that several of the datasets included were selected by stratified sampling designs, and these designs had intensified sampling in high burned area strata. According to the original articles associated with these datasets, rather complex estimation formulas have to be applied to such data (i.e., the less complicated formulas of simple random sampling are not appropriate when the sampling was stratified with different sampling intensities in the strata). It would therefore seem necessary that users of these reference datasets be cautioned about the need to use proper estimation formulas if users are to correctly report accuracy from these stratified sample datasets. This would also create the need to include in the datasets the information required to apply these estimation formulas, for example stratum sizes, the stratum ID of each sampled unit, and perhaps additional information depending on the specific details of the particular dataset.

Thank you for this observation. We have added the required data to use the validation datasets obtained through SRS to make probabilistic estimations of accuracy. The stratum ID of each sampled unit and the total area of the TSAs from which reference data was obtained have been added to the .csv files provided in the metadata folder. In addition, a table with the stratum sizes for each reference dataset is also provided in section 5 (Appendix A: Supplementary tables).

Technical Corrections and Suggestions

Throughout, readability would be enhanced by using paragraph indents at the start of each paragraph.
*Preprint manuscript was prepared according to the journal template, post-editing will be applied to the final version.*

L23: insert "a" to revise to "requires a high level"
*Done.*

L26, L29: Given that the acronym BARD was defined at Line 26, replace "The Database" with "BARD"
*Done.*

L40: "sensors" should be "sensor"
*Done.*

L41: revise to "reference data that are based on" ["data" is plural so "data that are"]
*Done.*

L46: "products" instead of "product efforts"
*Done.*

L63, L79, L105, L106, L159, L161, L164, L198, L205, L207, L209: Throughout the manuscript, the words "file" and "files" are sometimes used to the refer to the actual reference data. For example, at L63, the "files" were not derived from pairs of images, but rather the "reference data" that are stored in the files have been produced from the pairs of images. The text should be revised to replace "files" with "reference data" unless the text is referring to the actual files that store the reference data.
*Thank you for this observation, we have revised the document and changed it as suggested.*

L64: Replace "without probabilistic meaning" by "that were not selected using a probability sampling design". It is not clear what "direct sampling" is. Is direct sampling convenience, purposeful, or other sampling without randomization?
*We have modified the sentence to 'Early validation exercises were subjected to a first stage validation, usually based on small samples of reference sites that were not selected using a probability sampling design, but rather by a purposeful or convenience selection based on data availability or expert knowledge to ensure diverse wildfire conditions were included in the sample'.*

For all examples at Lines 65-70, it appears that there was a rationale for why sites were selected (even if they were not selected by a randomized protocol). It would be useful to mention what purposeful selection criteria were used. The Roy and Boschetti example mentions sites selected to be spatially distributed across the landscape, so this is an example where the manuscript provides useful additional information regarding the purposeful selection criteria.
*Validation sites selection in Chuvieco et al. (2008) was based on Landsat and CBERS images donated by regional space agencies, when Landsat archive wasn't free open to the public. We have mentioned it in the corresponding paragraph.*

L70-71: If Boschetti et al. (2019) collected data for only a single year, does that qualify as a "full spatio-temporal validation"? It would be helpful to define what a "full validation" is in regard to time and space.
*We have removed the expression 'full spatio-temporal validation' to avoid confusion and changed the sentence to 'A recent study has provided a validation of the MCD64A1 product implementing*

*a probability sampling design and using Landsat-8 Operational Land Imager (OLI) images, but only for a single year (Boschetti et al., 2019)'*.

L88: insert "design" after "random sampling" to create "stratified random sampling design"
Done.

L89: Consider revising to: "Boschetti et al. (2016) extended the sampling design to include the temporal dimension of the sampling units."
Done.

L90: insert "the" between "allocate sample" and delete "a" from "example a stratified"
Done.

L91: insert "the" before "sample"
Done.

L94: replace "are" by "is" because "dimension" is a singular noun
Done.

L99: delete "a"
Done.

L106: Consider revising to: "The procedures implemented to obtain those burn patches are diverse, depending…"
Done, text has been modified as suggested.

L109-110: Consider revising to: "Parts of the scene that cannot be observed or interpreted because of clouds or sensor problems (i.e., Scan Line …"
Done, text has been modified as suggested.

L115: replace "such" by "each" and replace "like" by "such as"
Done.

L153: Are n=127 and n=131 the number of TSAs sampled? It is not clear what these numbers represent.
The numbers refer to the number of images interpreted from each sensor, 127 images from Landsat-5 and 131 from Landsat-7. We have changed the sentence to clarify this point. Please, note that numbers have been modified because we initially included some reference data that shouldn't be included in this dataset.
*'A total of 210 images from Landsat-5 TM (n=101) and Landsat-7 ETM+ (n=109) satellite sensors were used to retrieve BA perimeters'*.

L170: delete "to each sample unit" because this threshold is applied to all TSAs. That is, all TSAs are assigned to strata as part of the sample selection process. It is not just the sampled units that are assigned to strata.
Thank you for pointing that out, we've changed the text according to your observation.

L172: given that "proportional allocation" for stratified sampling is defined as the sample size in each stratum being proportional to the number of units in the entire study region belonging to that stratum, replace "applying a proportional allocation" by "applying a sample allocation".
Done.

L182: replace "in" with "of" and replace "days" with "day"
Done.

L185: It is not clear how the actual time period covered by these "long units" is defined.
The long sampling units are defined by multiple consecutive pairs of images (short sampling units, separated by 16 days or less) covering at least 100 days. We have clarified the concept of short and long units in section 2.2.

L186: Consider revising to: "Reference maps using long units concatenate consecutive 8-16 day maps (Fig. 5)."
This line has been removed as long unit reference data generation methodology is now explained in section 2.2.

L188: The 50 units are for fire CCI Africa compared to 100 units per year for FireCCI global?
The authors used different sampling intensities for Africa and global. The 50 (long) units for FireCCI Africa implied an effort in the generation of reference data similar to that for the 12 years of FireCCI global. In the former case, 1052 pairs of images were processed, and on the latter case 1200.

L189: replace "consists on" with "consists of" and replace "perimeters" by "perimeter"
Done.

L190: replace "units" by "unit" (2 cases) and "days" by "day"
Done.

L198: remove "A" before "systematic sampling"
Done.

L201: replace "the whole" with "all" and replace "was" with "were"
Done.

L203: "consecutively" should be "sequentially"
Done.

L209: "joined" should be "joint" and "by" should be "between"
BrFLAS dataset has been removed from BARD (please, see SC6 response).

L213: delete "the" before "77%"
BrFLAS dataset has been removed from BARD (please, see SC6 response).

L219: replace "scar samples" by "scars sampled"
BrFLAS dataset has been removed from BARD (please, see SC6 response).

L223: "days" should be "day"
BrFLAS dataset has been removed from BARD (please, see SC6 response).

L224: "pair" should be "pairs"
BrFLAS dataset has been removed from BARD (please, see SC6 response)

L228-229: Continue to use the same phrasing as at L180 and L207 to identify the stage of the reference dataset. The sentence structure at L180 and L207 is much easier to read.
BrFLAS dataset has been removed from BARD (please, see SC6 response).

L231-232: replace "generated to perform the validation of the BAECV" with "generate to validate the BAECV"

As we have renamed BAECV dataset to CONUS Landsat Burned Area, the sentence "generated to perform the validation of the BAECV" has been replaced by "generate to validate the Landsat Burned Area product".

L232: Move the text "Landsat Burned Area Essential Climate Variable" to before the first use of BAECV at Line 231.
As we have renamed BAECV dataset to CONUS Landsat Burned Area, the sentence has been changed to: '*The Landsat Burned Area reference dataset (Vanderhoof et al., 2017;2020) extends across the contiguous United States (CONUS) and was generate to validate the Landsat Burned Area product (Hawbaker et al., 2017;2020)*'.

L238: delete "A" before "systematic"
Done.

L239: the three values of n sum to 335 images not 336
Thank you, the error has been corrected.

L243: replace "…only two (pre and post-fire image…" by "…only two images (pre and post-fire)…"
Done.

L266-267: Continue to use the same phrasing as at L180 and L207 to identify the stage of the reference dataset.
Done.

L272: "wildfires" should be "wildfire"
Done.

L279: "were" should be "was"
Done.

L283: "postfire" should be "post-fire"
Done.

L284: "formers" should be "former"
Done.

L290-291: Continue to use the same phrasing as at L180 and L207 to identify the stage of the reference dataset.
Done.

L306: Consider changing "futures updates come to replace the lack…" with "future updates remedy the lack…"
Done, text has been modified as suggested.

##################################################################################

**Anonymous Referee #2**

General Comments

This manuscript describes the availability of a new dataset comprising a compilation of reference burned area data, which can be used for the validation of burned area products. The short

description paper outlines the methods used to standardise a number of different datasets into a common format, and a more detailed description on each one. It also gives an overview of why validation is necessary but not always readily available, which provides useful context.

Validation of burned area products is definitely lacking in the field, and this is a welcome contribution to the research area. I think it will be useful for many researchers working on fire and burned area. The methods are well-referenced, and are mostly clearly described, with the exception of a few points as outlined below. The data is readily available via the link provided in the text, and can be accessed immediately after completing a short form. The data appears complete.

We would like to thank you for your positive comments about the contribution of the present work.

Specific comments

Section 2.1 Selection of validation sites: This section comes across as a literature review of different methods, and I'm not sure what is actually being implemented in this paper from reading this section. Can you clarify in the paragraph (e.g. does each dataset use a different method?)

This point was also indicated by the anonymous referee # 1 in his/her specific comment (point 1). We have added an explanation in section 2.1 clarifying this aspect. Please note that table 2 of the reviewed manuscript summarizes the sampling method applied in each dataset.

At the end of the Introduction, the overview of the paper is a bit vague. I think this would benefit from a clearer outline of the structure, and a list of the datasets that are considered in this paper to give a better overview up front.

Thank you, we have mentioned at the end of the introduction the datasets included in BARD and clarify the project where they have been produced and the contents of the manuscript.

Line 220 – only data in June to October is considered for this dataset. This covers the main fire season in this region, but how are the fires outside of the fire season dealt with?

BrFLAS dataset has been removed from the database since it does not follow CEOS cal-val standards. Please, see short comment 6 and response.

Presumably the temporal length of the reference files is such that it covers multi-day burning. It is worth pointing this out in the text explicitly.

Reference data include all the fire perimeters occurred between the two dates of the Landsat images used to generate them. This is a standard practice in BA validation. We have added a comment at the end of the section 2.2. and modified figure 5 to clarify this point.

It would be useful to include some text describing how one might use all these different reference datasets in practise. Should they all be used together, and if so how should the range be accounted for?

Thank you for this relevant question. This question is also related to specific comments (point 2) from the anonymous referee # 1, general comments of referee # 3 and SC3 (point4). Datasets are not supposed to be used together, as they have been obtained from different methods, rather users can choose the datasets that best suits their needs. As suggested by referee # 1, we have added the data necessary to make probability estimates of accuracy for those datasets obtained through stratified random sampling (Tables in Appendix A of the reviewed manuscript).

How were these datasets selected? Are there any other datasets available that are not included here, or are these the only ones available? I suggest including some explanation of this in the text.

Yes, there are other datasets that have been produced by other authors (e.g. Boschetti et al. 2016;2019). We made a general announcement through the GOFC-GOLD Fire implementation team list of scientists working on BA products and to our network of fire scientist. The resulting

database includes files those that the authors were willing to share publicly and met the CEOS cal-val standards.

Are all the datasets related to FireCCI? It seems so from the description of the data via the link, but not in the paper.
Only the datasets with the 'FireCCI' word in its name were produced under the FireCCI project, the rest of the datasets come from others projects. We mention this in the introduction: 'These validation files were compiled from different international projects and years…'. In addition, we have added the project name of each dataset in Table 2.

Most modellers use NetCDF, if it would be nice if this format was considered for future releases.
Thank you, we will keep in mind your suggestion for future releases. We don't usually use the NetCDF format for the reference files, but users interested in such format can easily do the conversion from .shp to .nc with the open tool 'ncl_convert2nc' that can be downloaded from '*https://www.ncl.ucar.edu/Document/Tools/ncl_convert2nc.shtml*'.

Technical corrections
References to figures ("Fig.") throughout the text is sometimes with a space and sometimes without
Thank you, we've added a space in those where it was missing.

Line 49 – change to "acquired in the year 2000"
Done.

Line 182 - "consists of"
Done.

#################################################################################

**Anonymous Referee #3**

General Comments

This manuscript describes a first attempt at compiling a common database of burned area reference perimeters ("BARD") suitable for validating remotely sensed burned area data sets. While the goal of producing the BARD is laudable, I feel the result falls somewhat short in that the authors provide no guidance in how this dataset should be used in practice. While reprojected and vectorized into a common format, the six underlying validation data sets were not generated in an entirely consistent manner and vary significantly in terms of sampling strategy and minimum mapping unit and various other important respects. As a result, I feel it is essential for the authors to advise users how the database as a whole should be used. For example, should some or all of the individual data sets be merged, or should they always be used separately? If the latter, then any validation of a global data set will yield six different sets of validation results. How should these results be interpreted, especially for the individual data sets that overlap in space and time, such as FireCCI Africa and FireCCI Africa S2? Furthermore, the authors state that "The Burned Area Reference Database will be expanded with new reference files that are being produced in the FireCCI project and we encourage future contributions from the scientific community", but it is not clear how this plan can scale practically as the number of data sets grows.

We have now included some reflections and information (Tables A1-A4) on practical uses of the database. For further details, the reader is also referred to the articles where each dataset was first published. We consider this as a collection of BA reference datasets, not as a single one.

Therefore, is up to the user to select certain regions or periods to produce his-her accuracy estimates. The uncertainty of accuracy estimates should contextualize the discrepancies between validation results from several datasets (and same product and overlaps in time and space). Slight discrepancies are expected as any single dataset is observing a sample of reference data instead of the whole population. We have now provided additional data to compute those accuracy metrics, but this database can be used in several different ways by potential users. Some, for instance, may use certain datasets for training their algorithm and some others for validation. Obviously, we do not aim to convert the BARD in a standard validation source, but just to provide useful data for BA algorithm developers and modellers.

Specific Comments

L41: "they require generating global reference data that is based on higher-resolution sensors" Although I agree with this statement, it overlooks studies such as Roteta et al. (2019) who used 30-m Landsat images to validate a 20-m Sentinel-2 burned area product.
The Roteta et al. paper performed a stratified random selection of Landsat images for generating the reference perimeters to compare accuracy metrics of S-2 and MODIS BA products. A previous validation based on a systematic sample of S-2 MSI images gave similar results, so only the last validation was included in the paper. It is certainly more convenient to use higher resolution images for validation, but in this case it was decided to use the same validation dataset to make comparisons between coarse and medium resolution sensor products more fair. In addition, a statistically design sample based on high-resolution images (Planet) is very complex and costly, and when using them for BA validation have been done in a very qualitative way (Roy, D.P., Huang, H., Boschetti, L., Giglio, L., Yan, L., Zhang, H.H., & Li, Z. (2019). Landsat-8 and Sentinel-2 burned area mapping - A combined sensor multi-temporal change detection approach. Remote Sensing of Environment, 231, 111254.)

L59: Giglio et al. (2018) give a release date of mid 2008 for the NASA MCD45A1 product.
We have corrected the date.

L68: "The MCD64A1 Collection 5 was not formally validated" Giglio et al. (2009) performed an "accuracy assessment for three geographically diverse regions (central Siberia, the western United States, and southern Africa)" using 50 Landsat scenes. Is this not validation?
Giglio et al. (2009) selected three different areas to represent different ecological conditions to evaluate their algorithm and no probability design was applied. The authors provided only the producer's accuracy for the scenes previously selected but didn't report global accuracy estimates of the product.

L85/Section 2.1: The authors note the importance of sampling design and describe various important components of this process, but not all of the BARD data sets seem to have adopted the strategies described in this section. It would be helpful to note any deviations within BARD from the sampling strategy described here. The authors might perhaps also provide a brief summary of the CEOS-LPVS validation stages to help readers interpret the stage numbers mentioned later for the individual data sets (in Table 2, for example) in the context of sampling.
Section 2.1 aims to provide a general overview of the sampling design methodologies developed for burned area validation. The particular sampling design adopted for each dataset is specified in the correspondent description of the datasets in section 2.4 and summarized in table 2.
Thank you for the suggestion, we have provided a description of the CEOS-LPVS validation stages.

L158: "The FireCCI global 2008 dataset includes 129 reference data files" This number differs slightly from Padilla et al. (2014), who refer to "102 sampled pairs". Presumably additional scenes were added to that data set. This is worth mentioning since it would alert readers that the summaries and/or statistics provided in Padilla et al. (2014) do not necessarily apply to the FireCCI global (2008) distributed in BARD.

The sampled units of such dataset comprises 105 units and the correct reference for this dataset is Padilla et al. (2014, 2015), the rest of the reference files (24) shouldn't be included in the dataset. The dataset has been updated including only these 105 reference files, and the dataset description has been updated accordingly.

L195/Section 2.4.4: The 2016 FireCCI Africa S2 data set is not mentioned in either of the references cited in this section. Please add the correct reference or clarify that the data set has not been previously published.
This dataset was used to perform and initial validation of the FireCCISFD11 product but has not been published. We have indicated this situation in Table 2 where we provide the related publication of each dataset.

L208/Section 2.4.5: Rodrigues et al. (2019) mention a minimum mapping unit of 21ha. Does this threshold also apply to the BrFLAS Brazil data distributed in the BARD?
No, no minimum mapping unit was applied to the BrFLAS Brazil. In any case, this dataset has been removed from the BARD since it does not follow CEOS cal-val standards. Please see short comment 6 and response.

L230/Section 2.4.6: Hawbaker et al. (2020) include the following remark about the BAECV validation data set: "Because no independent reference data were available for burned areas in agricultural cover types, the Landsat-based BAMS reference dataset did not train on agricultural fires and consequently cannot be considered accurate for this cover type." Have the unreliable reference polygons belonging to this category been flagged or removed from BARD? If not, some guidance to users about how they should identify and handle such cases would be appropriate.
The CONUS Landsat Burned Area (previously named BAECV) reference dataset classifies agriculture cover types as burned/unburned. The comment in Hawbaker et al. (2020) was made to acknowledge that because we lacked ancillary datasets in agriculture areas, the reference dataset burn classifications were not explicitly trained using agricultural burned polygons, and therefore, the reference dataset may be less accurate in this cover type. As 19 of the 28 TSAs contain at least some agricultural area it does not make sense to remove these shapefiles, however, in response to this comment we have added a sentence in the description of this dataset of the reviewed manuscript:

'…The low-, medium- and high-intensity development classes (i.e. urban areas) were masked using the National Land Cover Database (NLCD, https://www.mrlc.gov/national-land-cover-database-nlcd-2016) (Homer et al., 2015) to reduce spectral confusion between burned areas and impervious surfaces. *Similarly, agricultural burns were not used to train the reference data burn classification, therefore the accuracy of the reference dataset in agricultural areas is unknown. If this is of concern to users, then users can mask the "cultivated crops" land cover type from the reference data using the NLCD* '.

L242: "The pre- and post-fire image pairs did not specifically represent a probability sample within a year but were designed to target changes incurred over the peak fire season." Given this targeting of the peak fire season, is it appropriate to use this dataset for assessing out of season commission errors?
According to FireCCI51, the main peak fire season for CONUS goes from July to September-October. 80.36% of reference files from CONUS Landsat Burned Area dataset include months out of the fire season. Thus, we consider that this dataset is appropriate to assess Ce out of fire season.

L268/Section 2.4.7: Given that the NOFFi-OBAM mapping service "is activated after large wildfires events and under explicit requests by the local forest offices", is it appropriate to use this data set for assessing commission errors? Please explain and include appropriate caveats if necessary.

Yes, NOFFi-OBAM is appropriate for assessing commission errors as reference data follow CEOS cal-val standards. As we explain in the dataset description: 'The NOFFi-OBAM fire perimeters were used as basis for creating the reference data for the NOFFi Greece reference dataset' and we mention that 'Small fires within the specific time series that were not mapped from the NOFFi-OBAM service were explicitly digitized'. Additionally, unburned and unobserved categories were added to adapt this product to the CEOS cal-val standards.

Figure 3: This figure shows perhaps a dozen validation sites that are not shown in the equivalent figure of Padilla et al. (2014), where the 2008 FireCCI global validation dataset was originally described. Please see related L158 comment above.
The figure has already been corrected according to L158 response.

Figure 5 would be much more useful if it included clouds or some other source of missing data in the Landsat image stack. The long unit sampling is not clearly described in the manuscript, but I think I understand most of what the authors here poorly describe only after consulting Figure 12 of Padilla et al. (2018). Perhaps the authors could include a similar figure here.
Thank you, we have modified figure 5 (now figure 3) to clarify the schematic process to obtain long unit reference data. We also have extended the explanation on how long units are obtained in section 2.2.

Figure 9: Not clear why it is useful to highlight FireCCI TSAs vs. Added TSAs on the map. It would be more useful and more consistent to show the time period be-tween Landsat image pairs as was done for the other data sets in Figures 3, 6, 7, and 8.
The CONUS Landsat Burned Area dataset used 28 validation sites that were repeatedly sampled in each of the six validation years (with different time gaps in each year), making it challenging to provide a figure similar to Figures 3, 6, 7, and 8. This is the only dataset that was created to validate a specific region (CONUS) based on a previous existing global dataset (FireCCI global 2008) and this is a relevant aspect we mention in the reviewed manuscript:
'*another key advantage of stratified random sampling design that should be strongly emphasized is that it makes it possible to increase the sample size of an initial global sample for specific regions or rare land-cover classes (Stehman et al., 2012). This is the case of CONUS Landsat Burned Area (1988-2013) dataset where reference sites for the CONUS extent were augmented based on the initial sample of the FireCCI global (2008) dataset.*'. Figure 9 emphasize this property. In response to this comment we have added additional text to the Figure caption to clarify:
**"Reference data were generated for each TSA in each of the six sample years (1988, 1993, 1998, 2003, 2008, 2013)."**

Table 2: Please show the total areas of the separate burned, unburned, and no-data classes for each data set.
We have added this information in a separate table (table 3) as suggested in SC1.

BARD                DOI                landing                page (https://edatos.consorciomadrono.es/dataset.xhtml?persistentId=doi:10.21950/BBQQU7).    The landing page describes BARD almost exclusively as a FireCCI effort. This is a little bit inconsistent with the manuscript, which says that the database "was created by compiling existing reference burned area datasets from different international projects."
Yes, BARD is an initiative that arises from the FireCCI project and 92% of reference files were produced in the FireCCI project. However, we consider essential the present and future contributions of other initiatives to this effort.

Technical Corrections

L40: change "sensors" to "sensor"
Done

L57-58: Acronyms MERIS and MODIS not defined
Done.

L85: change "amount" to "number"
Done.

L91: change "sample" to "samples"
Done.

L213: change "covering the 77%" to "covering 77%"
BrFLAS dataset has been removed from the database since it does not follow CEOS cal-val standards. Please, see short comment 6 and response.

Figure 2 caption: change "Time distance between" to "Time period between"
Done.

#############################################################################

**David Roy short comment (SC1)**

Table 2 summarizes the number of reference files but does not provide summary information on the total areas of the 3 mapped classes (burned, unburned, no data). Please add a new table providing this information for each project and for all the projects together. This is needed because (i) Landsat and Sentinel-2 images have different areas (_185x185 km and _110x110 km), (ii) different image spatial subsets were mapped (i.e., not the entire image extents) by the different projects, (iii) the "no data" class includes areas where the interpreter did not undertake the mapping and it is unclear if this was a major proportion and/or if it varied among the projects. It would be helpful to then place the total mapped burned and unburned areas in the context of (a) the total global land area and (b) the typical total annual global area burned, and update the abstract and conclusion accordingly.

Thank you for your suggestion. Information about the total area from the 3 mapped classes was already included in the medatada files, however, we didn't mention it in the manuscript. We have added the proper comment in the .csv metadata file description and included the suggested table summarizing the total area mapped of each dataset and the area of the three mapped categories.

#############################################################################

**David Roy short comment (SC2)**

Could you explain the long and short units methodology better please. Currently as written I cannot understand it. Figure 5 is helpful but it does not show the case where there are "no data" areas (for example, due to cloud and/or the Landsat SLC-off issue) in the image time series. Please clarify this in the paper text - paying particular attention to how the "no data" pixels are handled in the long unit derivation. I suspect that there are underlying assumptions that reduce the utility of the long unit results for validation. For example, it is well known that in many regions the burn signal dissipates rapidly and that clouds occur commonly and often at the time of Landsat or Sentinel-2 overpass. Thus, to my mind, the long unit may (i) fail to capture the true area burned over the time series, (ii) reduce the proportion of the image area that is mapped as burned and/or unburned. Please clarify and discuss.

We have extended the description of the methodology to create the short and long units in section 2.2 and updated figure 5 (now figure 3) including unobserved areas, we hope it will be clearer now. As we explain in the methodology to build long units, consecutive pairs of images are used in order to avoid burn signal loss within the period covered by the long unit. On the other hand, it is true that may the proportion of the mapped region could be reduced in the spatial dimension, as 'no data' in any of the image pairs is kept into the final reference data. However, this should not affect the suitability of long units as reference data, please note that, for example, in Boschetti et al. (2019) images with cloud cover up to 70% are used for validation. Furthermore, long units have a crucial advantage over short units as they reduce the impact of the temporal reporting accuracy in the accuracy estimates. We consider that both, short and long units, are complementary and useful for validation.

###############################################################################

**L. Boschetti short comment (SC3)**

Given the effort and cost involved in generating validation dataset, the standardization and documentation of existing datasets for future use is certainly a meritorious effort, and there is no doubt that the datasets made available by the authors of this paper will find a use in the fire community.

I have however some concerns.
1) Scope of the paper and qualification of the dataset The way the dataset is presented could lead to some confusion and misinterpretation. The paper title refers to the 'development of a standard database' and the abstract refers to the present work as the compilation of 'the first Burned Area Reference Database'. This is misleading, because the work described in the present paper is limited to the collation of existing datasets, through standard GIS operations described in Section 2.3, namely the conversion of the various datasets to the same file format (shapefile), the use of standardized file names and the creation of simple metadata (Table 1). The wording of the abstract, i.e. referring to BARD as 'the first Burned Area Reference Database', is incorrect, as this is not the first burned area reference dataset - all the datasets that constitute the BARD are pre-existing. Maybe 'the first publicly available burned area reference dataset' would be a more appropriate statement.
We have followed your suggestion and changed the sentence to "the first publicly available burned area reference dataset". Actually, identical sentence was already included in the conclusion section: '*the first publicly available burned area reference dataset'* where we clearly mentioned that '*BARD is the first publicly available database that compiles and standardizes previously generated validation reference data.*'

2) Degree of novelty Section 2.3 is the only section that reports original work (i.e. the conversion of all data to shapefile, the standard filenames and the metadata), while the rest of the methods document what was done by the various research teams in the projects that provided the data.
The novelty of this paper is the compilation, standardization and public release of existing BA validation datasets, never done before.

3) Sampling. Section 2.1 ('Selection of the validation sites') describes a procedure for stratified random sampling of the burned area reference data that was followed by some of the collated datasets (but not all), which is extremely confusing. An inattentive reader might be led to believe that the BARD dataset itself is the result of a stratified random sampling, rather than the collation of datasets some of which were the results of stratified random sampling (albeit with different methods) and some that are not.
Table 2 and the documentation of the database included a clear description to the contents, but we have further clarified this point in section 2.1 to avoid confusion. Please, note that table 2 summarizes the sampling methodology applied to each dataset.

4) Stage 3 validation data set. Much is said throughout the paper of the compliance with CEOS Stage 3 validation requirements, but the BARD dataset as currently defined does not meet those requirements, i.e. it would not allow for the use of unbiased estimators of the accuracy metrics, and their associated standard errors. In the current form, pieces of BARD could be used for a Stage 3 validation, whereas other pieces could only be used for a Stage 1 or Stage 2 validation. Could the authors provide a harmonized statistical framework for the estimation of accuracy metrics from the whole BARD dataset?

BARD is a compilation of datasets that have been produced in different projects where different methods were applied. Even those datasets produced in the throughout the life of the FireCCI project present substantial differences. Please note that FireCCI project is a long term project that started in the year 2010 and, through the years, methods have been improving. That said, the aim of BARD is not to provide a harmonized statistical framework for all contributing datasets, because BARD is not a dataset itself but a compilation of datasets produced by different international projects and years. If that was the aim of BARD, we would only have made available the FireCCI global (2003-2014) dataset which is the one that covers the longest period but, instead, we choose to make all possible datasets available, and leave users the freedom to use the dataset or datasets that best suits their needs.

##################################################################################

**L. Boschetti short comment (SC4)**

Dataset harmonization It is apparent that the individual datasets collated in BARD were derived using a variety of semi-automatic procedures, and in the context of projects that had a somewhat different emphasis. How were the data harmonized so that they can be used together meaningfully? The title, abstract, introduction and conclusion imply a degree of harmonization between the datasets that is well beyond what was done, and might be seen as overstating the potential of the BARD to be a 'standard database'. Furthermore, there is no formal discussion in the paper of what requirements/ criteria/standards should be met by a 'standard database'.

We have answered this comment in point 4 of SC3. The standardization refers to the formats and documentation of the contributing datasets. This is something similar to what has been done in other databases made publicly available, as it is the case of a recent paper by Yebra et al. (2019) with fuel moisture content measurements.

##################################################################################

**Vitor Martins short comment (SC5)**

The standard datasets are relevant for validation of burned areas and the scientific community will be interested in such product. However, I found quality issues that limit the application without additional evaluation of files. Since these datasets have certain degree of automation in the production, further inspection is required to guarantee the high-quality in reference files. For instance, some areas present no-data/unobserved labels without a clear reason. When the reference dataset omits complex burned areas, the validation results tend to be biased. Other problems were observed in water pixels labeled as "unburned", and harvest areas as "burned". The authors should acknowledge the problems in these reference files and improve the quality as much as possible.

As burned perimeters of reference files were retrieved using semi-automatic classifications technics, all the reference files were visually inspected by an experienced interpreter and double checked (or triple checked in the case of the BAECV CONUS

(1988-2013) dataset) by another independent interpreter and the errors detected in the initial classification were manually edited until no errors were found. We mentioned this procedure throughout the manuscript in lines 76, 107, 177, 201-204, 218, 258 and 294. There are two main situations where you can find 'no-data/unobserved labels without a clear reason': the first one, is when the pre- or post-fire image (or both of them) used to retrieve the burned area has a region covered by clouds and shadow clouds, and the cloud/shadow mask applied to the images does not properly mask them. This situation makes difficult the correct classification of the pixels located near the clouds and could lead to an incorrect classification. To avoid this issue, cloudy regions of the image are excluded by using a mask manually created when necessary. This does not imply a reduction of the quality of the reference perimeters but a reduction of the interpreted area. The second one, is related mainly with crop areas, where harvested areas could be classified as burned areas as you point out in your comment. In some regions, especially those with dark soils, is very hard to differentiate between harvested crops and burned areas. Despite this issue, we made a great effort to interpret those areas and used some strategies to minimize the errors in those cases. In this sense, active fires from MODIS and observable flames on the images can help to identify which crops have burned and use only those pixels to train the classifier. However, there are situations where the classification results are quite uncertain and masking those areas as unobserved is preferable. Respect the water pixels labelled as 'unburned' issue, it has to do more with the established criteria in the validation methodly and not with labelling errors. Masking water as unobserved or no data could hide commission errors of coarse resolution BA products, especially in those regions where a large number of small water bodies surface cover a significant part of the validation area (e.g. Boreal and Tundra biomes). Labelling water as unburned was the criterion adopted in the FireCCI datasets and others reference datasets of BARD, we are aware that may this criterion is not the optimal for all the regions of the world but this is a question that requires further research to know exactly the impact of such decisions.

We acknowledge that reference files will have always a certain degree of uncertainty due to the remote sensing limitations but we consider that the reference files compiled in BARD are the best approximation to the ground truth that allows the current technology.

###########################################################################

**Vitor Martins short comment (SC6)**

I examined all the Brazilian (BrFLAS) data including comparing them to the multi-date Landsat images they were derived from. Two obvious issues:
1) None of the Brazilian data have a "no data\unobserved" class. This would only be correct if the images were always cloud- and shadow- free and but this is not the case.
For example, see below.
2) There are burned areas that are not mapped as "burned" because one of the images was cloud/shadow obscured. However, incorrectly, they have not been mapped as "no data\unobserved" (for example, see in red circle below). This makes these data difficult to use for validation, or as a reliable source of training data for classification purposes (as without looking at the images I would assume incorrectly that these areas were unburned).

Thank you very much for show us this issues about the BrFLAS dataset, we really appreciate the time dedicated to check the accuracy of the data. We agree with your

observations about the BrFLAS dataset and we have decided to exclude this dataset from BARD until these issues are fixed.

[revised manuscript text omitted]

---

## Editor Decision (ED1)

Dear Magi and co-authors,

many thanks for the very good revision of the manuscript that is ready for publication after addressing some small technical issues related to the references list

**(1) please add the actual version number to the reference of the data (this is relevant to relate the paper to exactly the version you are describing in the paper and change "doi:" to "https://doi.org/" (blue in the example below)**

Franquesa, M., Vanderhoof, M. K., Stavrakoudis, D., Gitas, I., Roteta, E., Padilla, M. and Chuvieco, E.: BARD: a global and regional validation burned area database, V.4.0, https://doi.org/10.21950/BBQQU7, 2020.

**(2) I have found some DOIs that were missing in your references list. Please add the following DOIs to the respective citation:**

- Broschetti et al. (2008): please add "https://doi.org/10.1029/2008JG000686"
- Cohen et al (2020): please add "https://doi.org/10.1016/j.rse.2010.07.010"
- Kennedy et al (2010): please add "https://doi.org/10.1016/j.rse.2010.07.008"
- Roy and Boschetti. (2008): please add "https://doi.org/10.1109/TGRS.2008.2009000" (and I would be delighted if you wrote the journals name IEEE Transactions on Geosciences and Remote Sensing, i.e. without using capital letters only)
- Tansey et al. (2003): Please add "https://doi.org/10.1029/2003JD003598"

**(3) Please preset all DOIs with the https://doi.org code**

- Chuvieco et al. (2008 and 2018), Eidenshink et al. (2007), Humber et al. (2019) and others only have the DOI number (beginning with "10."), please add https://doi.org/ before the 10.
- Please change all "doi:" to "https://doi.org/" in the reference list

Many thanks, also for choosing ESSD, and best regards,

Kirsten Elger

---

## Author Response (AR2)

**Response to the Topical Editor's comments**

Dear Topical Editor:

We would like to thank you for your positive comment on the revised manuscript. We have addressed all the final technical issues.

Please, note that we have added a new affiliation of one of the co-authors and corrected the reference in line 433.

With our best regards,
Magí Franquesa, on behalf of the co-authors

Find below the answers to each specific comment:

(Sentences in black and blue color are the original comments from the Topical Editor and our responses are marked in red color)

**(1) please add the actual version number to the reference of the data (this is relevant to relate the paper to exactly the version you are describing in the paper and change "doi:" to "https://doi.org/" (blue in the example below)**

Franquesa, M., Vanderhoof, M. K., Stavrakoudis, D., Gitas, I., Roteta, E., Padilla, M. and Chuvieco, E.: BARD: a global and regional validation burned area database, V.4.0, https://doi.org/10.21950/BBQQU7, 2020.

We have added the number version to the reference and changed the 'doi' as suggested

**(2) I have found some DOIs that were missing in your references list. Please add the following DOIs to the respective citation:**

- Broschetti et al. (2008): please add "https://doi.org/10.1029/2008JG000686"
- Cohen et al (2020): please add "https://doi.org/10.1016/j.rse.2010.07.010"
- Kennedy et al (2010): please add "https://doi.org/10.1016/j.rse.2010.07.008"
- Roy and Boschetti. (2008): please add "https://doi.org/10.1109/TGRS.2008.2009000" (and I would be delighted if you wrote the journals name IEEE Transactions on Geosciences and Remote Sensing, i.e. without using capital letters only)
- Tansey et al. (2003): Please add "https://doi.org/10.1029/2003JD003598"

All the missing DOIs have been added to the references.

**(3) Please preset all DOIs with the https://doi.org code**

- Chuvieco et al. (2008 and 2018), Eidenshink et al. (2007), Humber et al. (2019) and others only have the DOI number (beginning with "10."), please add https://doi.org/ before the 10.
- Please change all "doi:" to "https://doi.org/" in the reference list

We have checked the references and changed as suggested.

**(4) Line 254: remove (n=209)**

Done

**(5) Figure 4 (FireCCI global (2008)).** Please correct the figure as requested by referee 3.

The figure has been replaced with the correct one as requested

[revised manuscript text omitted]

---

## Editor Decision (ED2)

Dear Magí and co-authors,

many thanks for the very good revision of the manuscript that is ready for publication after addressing some small technical issues summarized in the attached document.

**(1) please add the actual version number to the reference of the data (this is relevant to relate the paper to exactly the version you are describing in the paper and change "doi:" to "https://doi.org/" (blue in the example below)**

Franquesa, M., Vanderhoof, M. K., Stavrakoudis, D., Gitas, I., Roteta, E., Padilla, M. and Chuvieco, E.: BARD: a global and regional validation burned area database, V.4.0, https://doi.org/10.21950/BBQQU7, 2020.

**(2) I have found some DOIs that were missing in your references list. Please add the following DOIs to the respective citation:**

- Broschetti et al. (2008): please add "https://doi.org/10.1029/2008JG000686"
- Cohen et al (2020): please add "https://doi.org/10.1016/j.rse.2010.07.010"
- Kennedy et al (2010): please add "https://doi.org/10.1016/j.rse.2010.07.008"
- Roy and Boschetti. (2008): please add "https://doi.org/10.1109/TGRS.2008.2009000" (and I would be delighted if you wrote the journals name IEEE Transactions on Geosciences and Remote Sensing, i.e. without using capital letters only)
- Tansey et al. (2003): Please add "https://doi.org/10.1029/2003JD003598"

**(3) Please preset all DOIs with the https://doi.org code**

- Chuvieco et al. (2008 and 2018), Eidenshink et al. (2007), Humber et al. (2019) and others only have the DOI number (beginning with "10."), please add https://doi.org/ before the 10.
- Please change all "doi:" to "https://doi.org/" in the reference list

**(4) Line 254: remove (n=209)**

**(5) Figure 4 (FireCCI global (2008)).** Please correct the figure as requested by referee 3.

Many thanks, also for choosing ESSD, and best regards,

Kirsten Elger